# RSPO3 impacts body fat distribution and regulates adipose cell biology in vitro

Nellie Y. Loh [1], James E. N. Minchin [2,3], Katherine E. Pinnick [1], Manu Verma [1], Marijana Todorčević[1], Nathan Denton [1], Julia El-Sayed Moustafa [4], John P. Kemp[5,6], Celia L. Gregson [7], David M. Evans [5,6], Matt J. Neville [1,8], Kerrin S. Small [4], Mark I. McCarthy[1,8,9], Anubha Mahajan[1,9], John F. Rawls [2], Fredrik Karpe[1,8 ✉] & Constantinos Christodoulides [1 ✉]

Fat distribution is an independent cardiometabolic risk factor. However, its molecular and cellular underpinnings remain obscure. Here we demonstrate that two independent GWAS signals at *RSPO3*, which are associated with increased body mass index-adjusted waist-to-hip ratio, act to specifically increase *RSPO3* expression in subcutaneous adipocytes. These variants are also associated with reduced lower-body fat, enlarged gluteal adipocytes and insulin resistance. Based on human cellular studies RSPO3 may limit gluteofemoral adipose tissue (AT) expansion by suppressing adipogenesis and increasing gluteal adipocyte susceptibility to apoptosis. RSPO3 may also promote upper-body fat distribution by stimulating abdominal adipose progenitor (AP) proliferation. The distinct biological responses elicited by RSPO3 in abdominal versus gluteal APs in vitro are associated with differential changes in WNT signalling. Zebrafish carrying a nonsense *rspo3* mutation display altered fat distribution. Our study identifies RSPO3 as an important determinant of peripheral AT storage capacity.

[1] Oxford Centre for Diabetes, Endocrinology and Metabolism, Radcliffe Department of Medicine, University of Oxford, Oxford OX3 7LE, UK. [2] Department of Molecular Genetics and Microbiology, Duke Microbiome Center, Duke University, Durham, NC 27710, USA. [3] Centre for Cardiovascular Science, University of Edinburgh, Edinburgh EH16 4TJ, UK. [4] Department of Twin Research and Genetic Epidemiology, King's College, London SE1 7EH, UK. [5] The University of Queensland Diamantina Institute, University of Queensland, Woolloongabba QLD 4102, Australia. [6] MRC Integrative Epidemiology Unit, Population Health Sciences, Bristol Medical School, University of Bristol, Bristol BS8 2BN, UK. [7] Musculoskeletal Research Unit, Translational Health Sciences, University of Bristol, Southmead Hospital, Bristol BS10 5NB, UK. [8] NIHR Oxford Biomedical Research Centre, OUH Foundation Trust, Oxford OX3 7LE, UK. [9] Wellcome Trust Centre for Human Genetics, Nuffield Department of Medicine, University of Oxford, Oxford OX3 7BN, UK. ✉email: fredrik.karpe@ocdem.ox.ac.uk; costas.christodoulides@ocdem.ox.ac.uk

Fat distribution is a heritable trait and an independent predictor of type 2 diabetes (T2D) and cardiovascular disease. While upper-body obesity is associated with adverse cardiometabolic outcomes, lower-body fat accumulation provides protection against T2D and atherosclerosis[1]. This is best exemplified by patients with partial lipodystrophy, a group of rare, heterogeneous disorders characterised by selective lower-body fat loss and development of severe insulin resistance[2]. The adverse consequences of truncal obesity are partly due to the anatomic location of visceral fat[3]. Complementarily, compared to subcutaneous (SC) abdominal (hereafter referred to as abdominal) and visceral, i.e., android adipose tissue (AT) depots, gluteofemoral AT displays differential fatty acid (FA) handling[1,3]. By favouring long-term FA storage, lower-body fat may protect extra-adipose tissues from ectopic fat deposition and lipotoxicity. Gluteofemoral AT may also promote systemic insulin sensitivity by secreting a more beneficial adipocytokine profile than android fat[4].

In children and adolescents, AT expansion occurs via an increase in both adipocyte number (hyperplasia) and size (hypertrophy). In contrast, in adults, abdominal AT expands by adipocyte hypertrophy while growth of lower-body fat might occur via hyperplasia[5,6]. Around one-tenth of the total adult fat cell pool is also renewed every year[5]. These findings highlight the importance of continuous adipocyte generation in the growth and maintenance of AT depots. Adipocytes originate from adipose progenitors (APs); mesenchymal stem cells and preadipocytes. It is now established that different fat depots arise from unique APs with distinct biological properties and developmental gene expression signatures[2,7–10], which give rise to specialised adipocytes with diverse functional traits. These findings have led to the hypothesis that developmental pathways play a critical role in establishing AP and adipocyte identity and thus in determining the size of fat depots, by modulating adipocyte number and size in each depot. Consistent with this theory, annotation of genome-wide association study (GWAS) meta-data revealed enrichment for developmental genes at loci influencing waist-hip ratio adjusted for BMI (WHRadjBMI), a surrogate measure of fat distribution[11]. Relevant to this work, genetic variants at the RSPO3 locus were shown to be the strongest genetic determinants of WHRadjBMI[12] although, the causal variants, cis-effector gene (s) and target tissue(s) at this locus have not been mapped.

WNTs are secreted glycoproteins that act locally, via multiple pathways, to regulate embryonic development and adult tissue homoeostasis[13]. In the best characterised, canonical cascade, WNT binding to their receptors leads to nuclear accumulation of the transcriptional co-activator β-catenin and increased WNT target gene expression. Besides WNTs, another group of secreted proteins, R-spondins (RSPO1-4) also amplify WNT pathway activity by engaging LGR4-6 receptors to increase the cell membrane density of WNT receptors[14]. Canonical WNT signalling plays a key role in stem cell and AP biology[13,15,16]. Accordingly, R-spondins potently stimulate the expansion of diverse adult stem cell types in vitro and in vivo and function as organ size regulators[14,17–19]. Lineage tracing studies have also identified LGR5 and LGR6 as exquisite adult stem cell markers in several adult mouse tissues[20]. LGR6 was additionally shown to mark mouse APs[21].

Prompted by these findings and the aforementioned GWAS meta-data[12] we investigated the molecular, cellular and whole-body effects of WHRadjBMI-associated alleles at RSPO3. Herein, we demonstrate that WHRadjBMI-increasing alleles at this locus act to increase RSPO3 expression specifically in SC AT. In human cellular studies, we further show that RSPO3 functions to modulate AP and adipocyte biology. Consistent with these findings, a nonsense rspo3 mutation is associated with altered fat distribution in zebrafish.

## Results

**Adipose-specific RSPO3-regulation mediates WHR associations.** GWAS results have identified associations between at least two independent signals at RSPO3 and WHRadjBMI[12]. Associations at both the sentinel (rs72959041) and secondary (rs1936807) single-nucleotide variants (SNVs) were stronger in females. We extended these findings by mining genotype and dual-energy X-ray absorptiometry (DXA) data from over 4500 adults from the Oxford Biobank (OBB) (Table 1). In the case of the secondary signal we used rs9491696 as a proxy for rs1936807 based on high mutual linkage disequilibrium (LD) ($r^2 = 0.89$). As the two association signals described are in perfect LD ($D' = 1$), data for both the index and secondary SNVs are shown with and without adjustment for rs9491696 and rs72959041 genotype respectively (Table 1). We found that the WHRadjBMI-increasing alleles at both rs72959041 and rs9491696 were associated with a redistribution of fat from the lower- to the upper-body, which appeared to be driven by a reduction in leg fat mass at both signals and a concomitant increase in android fat mass at rs72959041 (Table 1). Both signals were also associated with insulin resistant phenotypes based on data from the largest available GWAS meta-analyses for multiple traits (Supplementary Table 1). Notably, the association of the secondary signal as represented by rs1936807, with fasting insulin was stronger in women consistent with its sexually dimorphic association with fat distribution (male: $\beta = 0.0073$, $p$-value $= 0.06$; female: $\beta = 0.016$, $p$-value $= 3.2 \times 10^{-5}$, sex-difference $p$-value $= 3 \times 10^{-5}$) (data from the MAGIC Investigators, https://www.magicinvestigators.org/Lagou et al.).

We next sought to refine the location of the causal variants responsible for the GWAS associations at RSPO3 by constructing credible sets that collectively accounted for ≥99% of the posterior probability of association, using data from the GIANT consortium and UK Biobank combined meta-analysis[12]. The causal variant responsible for the primary WHRadjBMI association signal at RSPO3 was resolved to a single SNV, rs72959041 (Supplementary Data 1). Based on chromatin-state maps this variant maps to an AT enhancer element[22]. To determine if this signal was associated with a cis-expression quantitative trait locus (eQTL) in AT, we examined RSPO3 messenger RNA (mRNA) abundance in abdominal and gluteal fat biopsies from 200 adults (Supplementary Table 2). Compared with females homozygous for the common allele, female carriers of the WHRadjBMI-increasing allele at rs72959041 displayed higher RSPO3 expression in both fat depots (Supplementary Fig. 1a, b). We confirmed and extended this result by demonstrating allelic expression imbalance (AEI) in a subgroup of samples from seven female and seven male heterozygous subjects (Supplementary Table 3) using a tagging, 5′ UTR RSPO3 SNV (rs577721086, $r^2 = 0.98$ with rs72959041) (Fig. 1a, b). Robust cis-eQTL effects at rs72959041 were also detected in femoral AT in GTEx[23] and three independent studies of expression in abdominal AT (TwinsUK, METSIM[24], and deCODE[25]) (Supplementary Table 4). The cis association at rs72959041 was specific to SC AT, as no eQTL signal was observed in visceral fat or 44 other tissues in GTEx. Formal co-localisation analysis using coloc confirmed that the WHRadjBMI GWAS association at rs72959041 and the abdominal AT cis-eQTL were explained by the same causal variant (Fig. 1c and Supplementary Data 1). No other gene mapping within 1 MB of rs72959041 was associated with an eQTL in any tissue in GTEx or in abdominal AT from the METSIM cohort, indicating that RSPO3 is the likely mediator of the WHRadjBMI association at this signal. Finally, AEI analyses in fractionated abdominal and gluteal AT from ten females and three males established that the WHRadjBMI-increasing allele at rs72959041 was associated with increased RSPO3 expression in mature

**Table 1 Associations between RSPO3 SNVs and measures of body-fat distribution (DXA) in OBB subjects, adjusted for age and % total fat mass, and in combined analyses, for sex.**

| Measure | rs72959041 (adj. for rs9491696) EA = A, EAF = 0.01 | | | | | | rs9491696 (adj. for rs72959041) EA = G, EAF = 0.49 | | | | | |
|---|---|---|---|---|---|---|---|---|---|---|---|---|
| | Female (n = 2585) | | Male (n = 1958) | | All (n = 4543) | | Female (n = 2604) | | Male (n = 1967) | | All (n = 4571) | |
| | β | p | β | p | β | p | β | p | β | p | β | p |
| Android/gynoid fat ratio | 0.06 (0.05) | **$2 \times 10^{-6}$ ($3 \times 10^{-5}$)** | 0.04 (0.04) | **0.006 (0.02)** | 0.05 (0.04) | **$3 \times 10^{-8}$ ($8 \times 10^{-7}$)** | **0.004 (0.07)** | 0.04 (0.02) | 0.02 (0.01) | 0.1 (0.3) | 0.03 (0.02) | **0.0008 (0.04)** |
| Android/leg fat ratio | 0.06 (0.05) | **$7 \times 10^{-6}$ ($1 \times 10^{-4}$)** | 0.05 (0.04) | **0.002 (0.007)** | 0.05 (0.04) | **$2 \times 10^{-8}$ ($1 \times 10^{-6}$)** | **0.0008 (0.02)** | 0.04 (0.03) | 0.03 (0.02) | **0.046 (0.2)** | 0.03 (0.02) | **$8 \times 10^{-5}$ (0.007)** |
| Android fat mass (g) | 0.03 (0.03) | **$8 \times 10^{-5}$ ($3 \times 10^{-4}$)** | 0.01 (0.01) | **0.046 (0.055)** | 0.02 (0.02) | **$6 \times 10^{-6}$ ($3 \times 10^{-5}$)** | **0.04 (0.3)** | 0.01 (0.008) | 0.004 (0.0006) | 0.6 (1) | 0.01 (0.006) | **0.04 (0.3)** |
| SC android fat mass (g) | 0.03 (0.03) | **0.001 ($8 \times 10^{-4}$)** | −0.004 (−0.01) | 0.8 (0.4) | 0.01 (0.01) | **0.04 (0.08)** | **0.003 (0.01)** | 0.003 (−0.006) | 0.02 (0.03) | 0.055 (**0.04**) | 0.01 (0.008) | 0.1 (0.2) |
| Android visceral fat mass (g) | 0.03 (0.02) | 0.054 (0.2) | 0.03 (0.04) | **0.01 (0.009)** | 0.03 (0.02) | **0.004 (0.02)** | **0.003 (0.01)** | 0.04 (0.04) | −0.004 (−0.01) | 0.8 (0.4) | 0.03 (0.02) | **0.006 (0.04)** |
| Gynoid fat mass (g) | −0.02 (−0.01) | 0.08 (0.2) | −0.01 (−0.008) | 0.2 (0.4) | −0.01 (−0.01) | **0.04 (0.1)** | 0.09 (0.2) | −0.02 (−0.01) | −0.01 (−0.01) | 0.1 (0.2) | −0.01 (−0.01) | **0.02 (0.08)** |
| Leg fat mass (g) | −0.03 (−0.02) | **0.02 (0.08)** | −0.03 (−0.02) | **0.02 (0.07)** | −0.02 (−0.02) | **0.001 (0.01)** | **0.003 (0.01)** | −0.03 (−0.03) | −0.03 (−0.02) | **0.02 (0.06)** | −0.03 (−0.02) | **0.0002 (0.002)** |
| Total fat mass (g) | 0.009 (0.01) | 0.2 (0.1) | 0.004 (0.006) | 0.5 (0.4) | 0.008 (0.01) | 0.09 (0.052) | 0.5 (0.3) | −0.004 (−0.007) | −0.006 (−0.008) | 0.4 (0.3) | −0.004 (−0.007) | 0.4 (0.2) |

DXA dual-energy X-ray absorptiometry, EA effect allele, EAF effect allele frequency, OBB Oxford Biobank, SC subcutaneous, SNV single-nucleotide variant.
Significant p-values are given in bold.

adipocytes (Fig. 1d, e, Supplementary Fig. 1c, d and Supplementary Table 5). A weaker AEI signal was also observed in abdominal APs.

The causal variant responsible for the association at rs9491696 could not be resolved beyond a set of 13 SNVs in high mutual LD ($r^2 \geq 0.84$) spanning a ~ 27 KB interval (Supplementary Data 1). Two of these associated variants overlap enhancer histone marks in AT[22]. At this signal, the WHRadjBMI-increasing allele was not associated with elevated *RSPO3* expression in either abdominal or gluteal AT from both sexes after excluding carriers of the effect allele at rs72959041 from analyses (Supplementary Fig. 1e, f). However, using samples from a subgroup of 15 female and 17 male heterozygous subjects (Supplementary Table 6) we demonstrated robust AEI using a tagging, exonic *RSPO3* SNV (rs1892172, $r^2 = 0.84$ with rs9491696) (Fig. 1f, g and Supplementary Fig. 1g, h). A cis-eQTL effect at rs9491696 was also detected in abdominal AT from the TwinsUK study (Supplementary Table 4). No eQTL signal was evident in the skin or lymphoblastoid cell lines from the same TwinsUK subjects and no other gene localised within 1 MB of rs9491696 was associated with an eQTL in any tissue in this cohort or in GTEx. Formal co-localisation analysis confirmed that the WHRadjBMI GWAS association represented by rs9491696 and the abdominal AT cis-eQTL were explained by the same causal SNVs (Fig. 1c). AEI analyses in fractionated AT from 19 females (Supplementary Table 7) further established that the WHRadjBMI-increasing allele at rs9491696 was associated with increased *RSPO3* expression specifically in mature adipocytes (Fig. 1h, i). Weaker AEI signals were also detected in mature adipocytes and abdominal APs from 22 male subjects (Supplementary Fig. 1i, j and Supplementary Table 7). All individuals included in the AEI analyses where homozygous for the common allele at rs72959041. We conclude that the WHRadjBMI-associations attributable to the *RSPO3* locus are a consequence of increased AT *RSPO3* expression possibly as a result of sequence variation in adipocyte enhancers within intron 1 of *RSPO3*.

**Increased AT RSPO3 expression impacts adipocyte size**. We investigated the link between increased adipose *RSPO3* expression and AT morphology by histological assessment of abdominal and gluteal fat biopsies from 15 age- and BMI-matched pairs of females heterozygous for the WHRadjBMI-raising allele or homozygous for the ancestral allele at rs72959041 (Supplementary Table 8). This revealed that median adipocyte size was increased in the gluteal depot of WHRadjBMI-increasing variant carriers, due to a lower proportion of small adipocytes (Fig. 1j–m and Supplementary Fig. 2). No difference in abdominal adipocyte size was detected between the two genotypes despite the WHRadjBMI-increasing allele at rs72959041 being associated with increased android fat mass (Table 1).

**RSPO3 is expressed in a sex- and depot-specific manner in AT**. To gain mechanistic insights into how RSPO3 modulates fat distribution and adipocyte size we measured RSPO3 mRNA abundance in paired abdominal and gluteal fat biopsies from 200 adults. RSPO3 expression was sexually dimorphic, being higher in males and was greater in abdominal versus gluteal fat (Fig. 2a). RSPO3 mRNA levels were also higher in visceral compared to SC AT in both sexes (Fig. 2b). We additionally analysed RSPO3 expression in fractionated abdominal and gluteal AT from 61 females, 43 of whom had undergone a concomitant DXA scan to assess their fat distribution (Supplementary Table 9). RSPO3 mRNA abundance was greater in mature adipocytes versus APs and in abdominal versus gluteal adipocytes (Fig. 2c). Finally, we asked if upper- compared with lower-body fat distribution was

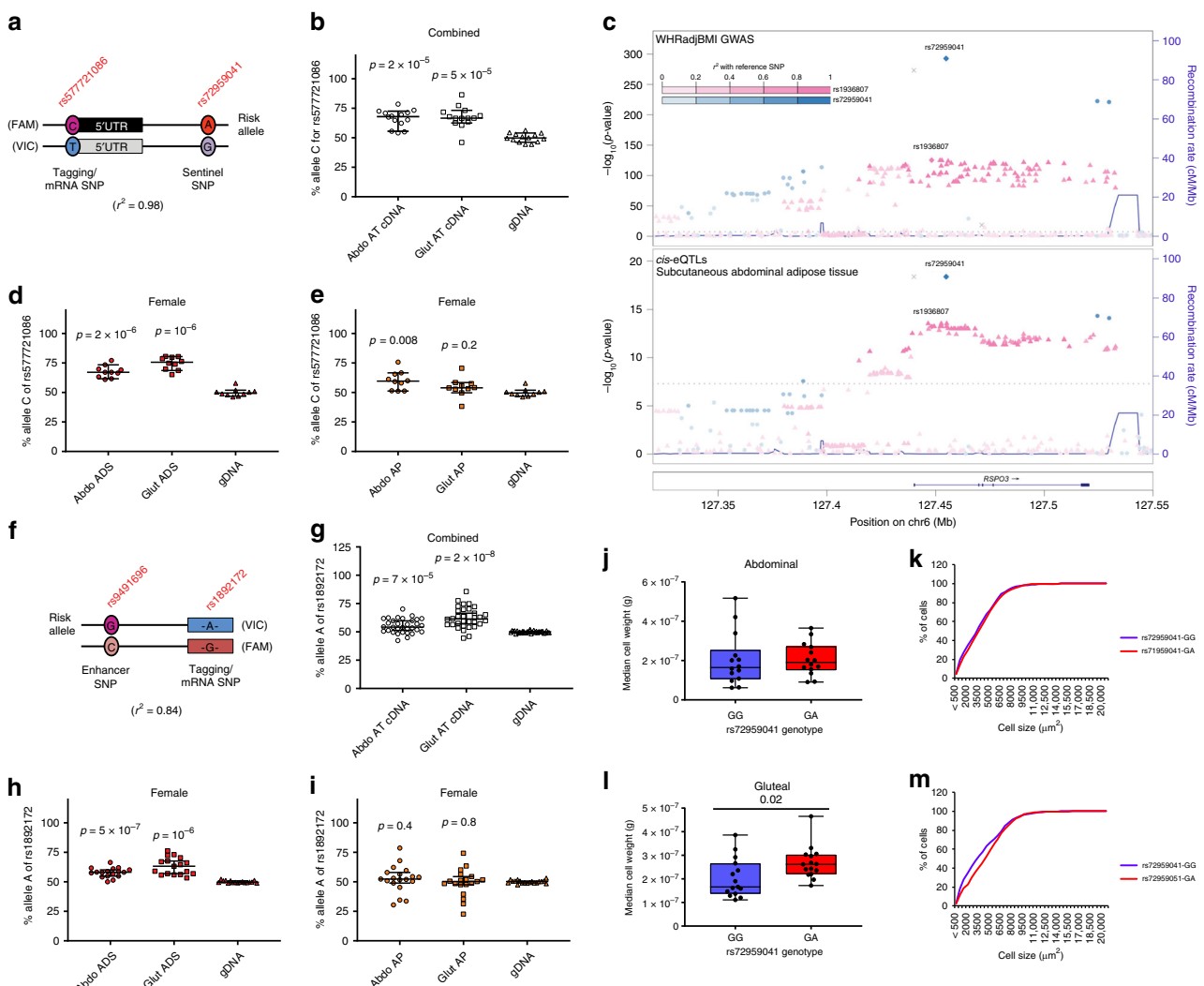

**Fig. 1 Allelic expression imbalance of *RSPO3* and histological studies in abdominal and gluteal AT. a** Schematic illustration showing alleles of the sentinel SNP rs72959041 and the tagging SNP rs577721086. **b** Allelic expression analysis of *RSPO3* transcripts was performed by qRT-PCR in abdominal and gluteal AT cDNA from 14 heterozygous carriers at rs72959041 (seven females, seven males). The proportion of total cDNA and gDNA containing the (WHR-increasing) rs577721086-C allele ($r^2 = 0.98$ with rs72959041-A) is quantified on the *y*-axis. Error bars are median values with 95% confidence intervals. Statistical significance was assessed by two-tailed paired Student's *t*-tests versus gDNA. **c** *RSPO3* WHRadjBMI GWAS signals co-localise with subcutaneous adipose cis-eQTLs. **d, e** Allelic expression of *RSPO3* transcripts were analysed in abdominal and gluteal isolated adipocyte (ADS) (**d**) and cultured AP (**e**) cDNAs from ten heterozygous female carriers at rs72959041 as in **b**. **f** Schematic illustration showing alleles of the enhancer SNP rs9491696 and the tagging SNP rs1892172. **g–i** Allelic expression analysis of *RSPO3* in **g** abdominal and gluteal AT cDNA from 32 individuals (15 females, 17 males) and in **h** abdominal and gluteal ADS and **i** cultured AP cDNA from 17 and 19 females, respectively, who are heterozygous at rs9491696. All individuals selected are homozygous for the rs72959041-G (non-risk) allele. The proportion of cDNA and gDNA containing the WHRadjBMI-increasing rs1892172-A allele ($r^2 = 0.84$ with rs9491696-G) is quantified on the *y*-axis. Error bars are median values with 95% confidence intervals. Statistical significance was assessed by two-tailed paired Student's *t*-tests versus gDNA. **j–m** Median adipocyte cell weight calculated from median cell area measured from **j** abdominal and **l** gluteal AT histological sections from 14 and 15 pairs, respectively, of age- and BMI-matched females grouped by rs72959041 genotype. Cumulative frequency distribution of the adipocyte cell-surface area are shown for abdominal (**k**) and gluteal (**m**) AT. More than 250 cells were measured for each biopsy. Box and whisker plot: centre line, median; box limits, upper and lower quartiles; and whiskers, maximum and minimum values. Statistical significance was assessed by a two-tailed Wilcoxon signed-rank test. Source data are provided as a Source Data file.

associated with differential AT RSPO3 expression. In age- and percent fat mass-adjusted partial correlations, gluteal AP RSPO3 expression correlated negatively with gynoid and leg fat mass (Table 2). No correlations between abdominal AP RSPO3 mRNA levels or mature adipocyte RSPO3 expression and fat distribution were detected in this cohort.

**RSPO3 modulates AP and adipocyte biology.** After mapping the expression profile of RSPO3 in human AT we investigated its role

in AP biology using immortalised APs generated from paired abdominal (imAbdo) and gluteal (imGlut) fat biopsies from a healthy adult male[26,27]. Stable RSPO3 knockdown (KD) in these cells (Fig. 3a) led to reduced imAbdo AP proliferation (Fig. 3b) and enhanced adipogenesis in both imAbdo and imGlut cells (Fig. 3c–f). Despite identical percentage KD (~ 45%) the pro-adipogenic effect of RSPO3-KD was more pronounced in imGlut versus imAbdo APs. RSPO3-KD in primary APs from a female subject recapitulated these findings although, adipogenesis was only augmented in gluteal cells (Fig. 3g–j). Finally, RSPO3-KD in

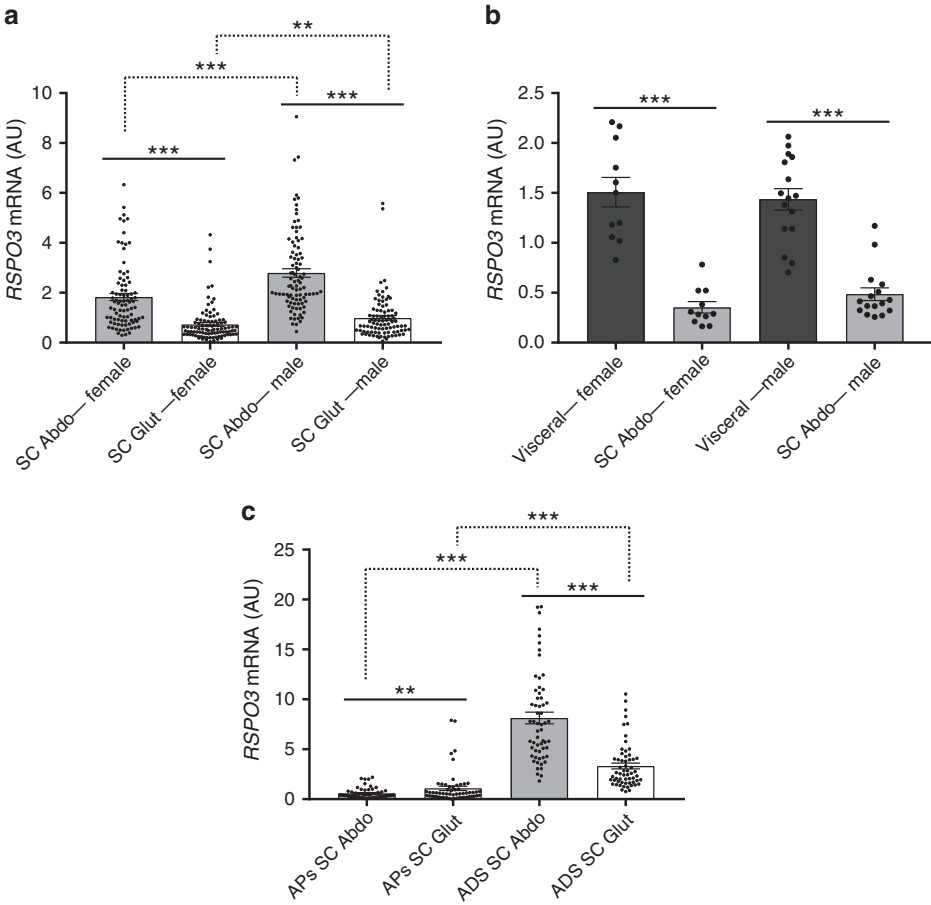

**Fig. 2 RSPO3 expression in human abdominal and gluteal whole AT and AT fractions.** Normalised RSPO3 mRNA levels in: **a** paired SC abdominal (Abdo) versus gluteal (Glut) fat biopsies from healthy women ($n = 103$) and men ($n = 97$), **b** paired visceral versus SC abdominal AT from 11 women and 16 men, and **c** fractionated AT [cultured APs and isolated mature adipocytes (ADS)] from 59 women. Histogram data are means ± s.e.m. **$p < 0.01$, ***$p < 0.001$, statistical significance was assessed by univariate analyses adjusted for age and BMI, and with Bonferroni correction for multiple testing for **a**, and by two-tailed paired Student's *t*-tests for **b** and **c**. Source data are provided as a Source Data file.

**Table 2 Partial correlations (Spearman's) of measures of body-fat distribution (DXA), with *RSPO3* mRNA levels from abdominal and gluteal adipose tissue fractions from 43 women, adjusted for age, % total fat mass and menopause status.**

| Traits | SC abdominal APs | | Gluteal APs | | Abdo ADS | | Glut ADS | |
|---|---|---|---|---|---|---|---|---|
| | rho | p | rho | p | rho | p | rho | p |
| Android/gynoid fat ratio | 0.143 | 0.4 | 0.331 | **0.04** | −0.126 | 0.4 | −0.071 | 0.7 |
| Android/leg fat ratio | 0.105 | 0.5 | 0.290 | 0.07 | −0.121 | 0.5 | −0.052 | 0.7 |
| Android fat mass (g) | −0.036 | 0.8 | −0.186 | 0.3 | 0.123 | 0.4 | 0.172 | 0.3 |
| SC android fat mass (g) | −0.038 | 0.8 | −0.339 | **0.03** | 0.075 | 0.6 | 0.044 | 0.8 |
| Android visceral fat mass (g) | 0.099 | 0.6 | 0.216 | 0.2 | 0.004 | 1 | 0.104 | 0.5 |
| Gynoid fat mass (g) | −0.137 | 0.4 | −0.441 | **0.005** | 0.207 | 0.2 | 0.131 | 0.4 |
| Leg fat mass (g) | −0.121 | 0.5 | −0.440 | **0.005** | 0.139 | 0.4 | 0.102 | 0.5 |
| Tissue android, %fat | 0.134 | 0.4 | 0.201 | 0.2 | −0.115 | 0.5 | 0.033 | 0.8 |
| Tissue gynoid, %fat | −0.320 | **0.047** | −0.355 | **0.03** | 0.162 | 0.3 | −0.029 | 0.9 |
| Tissue leg, %fat | −0.174 | 0.3 | −0.257 | 0.1 | 0.096 | 0.6 | −0.117 | 0.5 |

*APs* adipose progenitors, *ADS* adipocytes, *DXA* dual-energy X-ray absorptiometry, *SC*, subcutaneous.
Significant *p*-values are given in bold. Source data are provided as a Source Data file.

primary visceral progenitors led to enhanced adipogenesis and impaired proliferation (Supplementary Fig. 3). In reciprocal gain-of-function experiments, treatment with recombinant human RSPO3 stimulated proliferation in primary abdominal APs while inhibiting adipogenesis, more robustly in primary gluteal cells (Fig. 3k–m). The anti-adipogenic effect of RSPO3 was more pronounced in the presence of foetal bovine serum (FBS), which has been shown to promote WNT secretion from cultured cells[28,29]. Consistent with the phenotypic effects of RSPO3-KD, KD of LGR4, the predominant RSPO receptor expressed in AT (data from GTEx), impaired proliferation selectively in primary abdominal APs while stimulating adipogenesis more robustly in

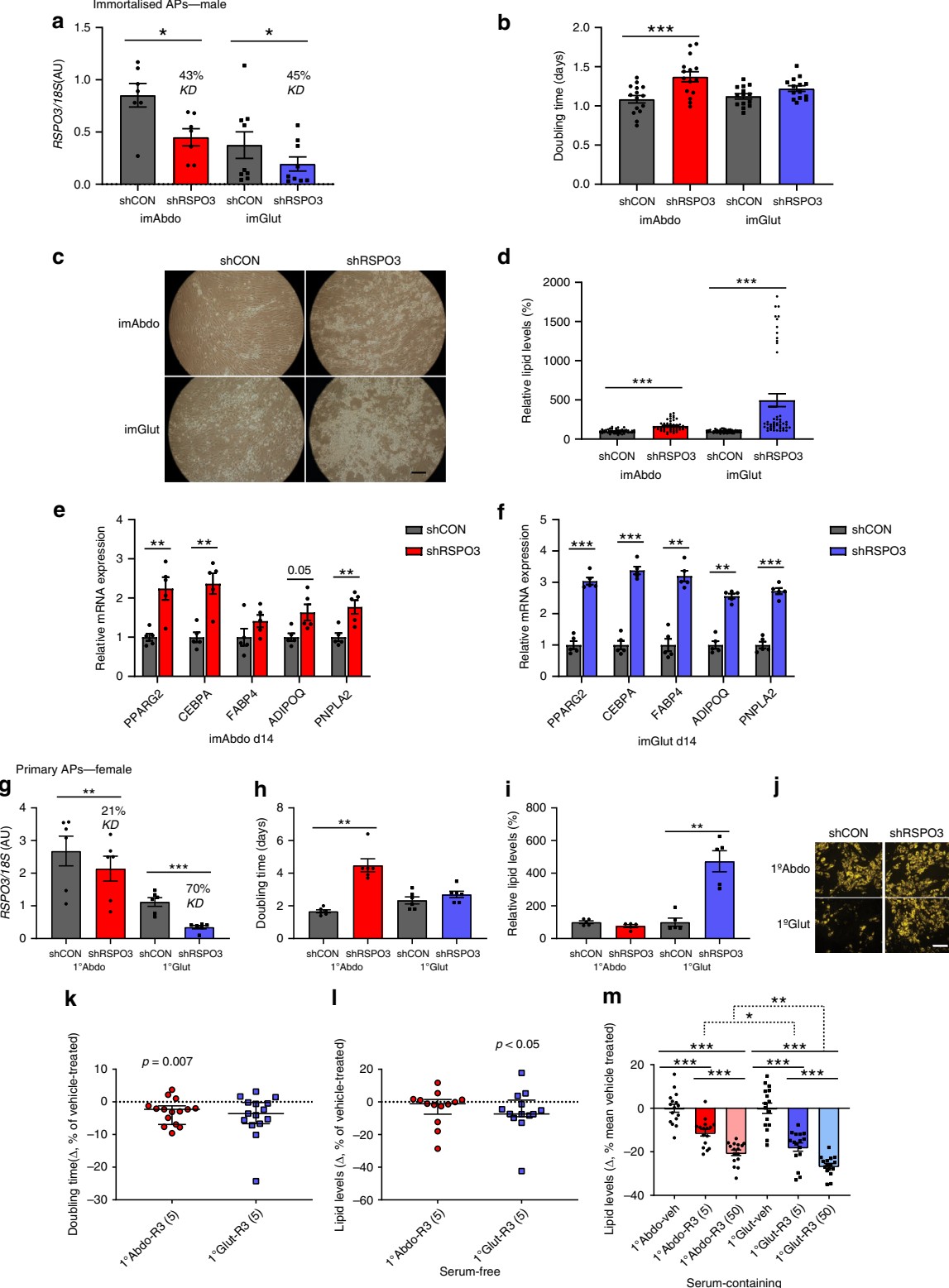

gluteal cells (Supplementary Fig. 4). We conclude that in vitro RSPO3 appears to primarily inhibit gluteal AP differentiation while stimulating proliferation of abdominal APs.

Given that WHRadjBMI-associated variants at RSPO3 act primarily to modulate RSPO3 expression in mature adipocytes we also examined the effects of induced RSPO3-KD on adipocyte function, using de-differentiated fat (DFAT) cells generated from

imAbdo and imGlut adipocytes. These retain their distinct cellular identities and many of their depot-specific gene expression signatures[30] (Supplementary Figs. 5 and 7a) and have a higher adipogenic potential compared to primary and immortalised APs (Fig. 4a). We focused initially on glucose uptake and lipolysis since the insulin signalling and intracellular lipolysis pathways have been genetically linked to the regulation

**Fig. 3 Effects of *RSPO3*-KD and recombinant human RSPO3 (rhRSPO3)-treatment on SC abdominal and gluteal AP biology. a–f** Effects of RSPO3-KD in immortalised (im) APs. **a** RSPO3-KD was confirmed by qRT-PCR (Abdo, $n = 7$ experiments; Glut, $n = 9$ experiments). **b** RSPO3-KD results in increased doubling time in imAbdo but not imGlut APs. Doubling time of control (shCON) and RSPO3-KD (shRSPO3) imAbdo ($n = 15$ experiments) and imGlut APs ($n = 14$ experiments). **c–f** RSPO3-KD promotes adipogenesis more robustly in imGlut versus imAbdo cells. Representative micrographs of shCON and shRSPO3 imAbdo and imGlut cells after 14 days of differentiation are shown (**c**). Scale bar = 500 μm. Adipogenesis was assessed by **d** AdipoRed staining (four independent experiments, $n = 12$ replicates each, expressed as relative lipid levels), and **e–f** qRT-PCR of adipogenic genes ($n = 5$ experiments). **g–j** Effects of RSPO3-KD in primary (1°) APs. (**g**) RSPO3-KD was confirmed by qRT-PCR ($n = 6$ experiments). RSPO3-KD leads to increased doubling time in 1°Abdo APs ($n = 6$ experiments) (**h**) and enhanced adipogenesis in 1° Glut cells (five independent experiments) (**i**). Representative micrographs of AdipoRed-stained control and shRSPO3 cells at differentiation day 12 are shown (**j**). Intracellular lipids are stained yellow. Scale bar = 200 μm. **k–m** Effects of rhRSPO3 treatment on 1° AP (**k**) proliferation ($n = 15$ [cells from three female subjects]), and **l–m** adipogenesis under **l** serum-free conditions ($n = 13$ independent experiments [cells from seven female donors]) and **m** in the presence of serum (FBS) ($n = 2$ independent experiments, eight replicates each). veh., vehicle; R3 (5), 5 ng ml$^{-1}$ rhRSPO3; R3 (50), 50 ng ml$^{-1}$ rhRSPO3. (**a, b, d–i, m**) Histogram data are means ± s.e.m. **k, l** Error bars are median values with interquartile ranges. *$p < 0.05$, **$p < 0.01$, ***$p < 0.001$. Statistical significance was assessed by two-tailed Student's $t$-test (**a, b, e–i**, paired; **d, m**, unpaired, and with Bonferroni correction for **m**) and a two-tailed Wilcoxon signed-rank test (**k, l**). Source data are provided as a Source Data file.

of fat distribution[11,31,32]. Forty-eight-hour doxycycline treatment led to efficient RSPO3-KD in both abdominal and gluteal in vitro differentiated DFAT adipocytes without affecting lipid accumulation or adipogenic gene expression (Fig. 4b, c and Supplementary Fig. 6). Adipocyte RSPO3-KD was not associated with changes in basal or insulin-stimulated glucose uptake (Fig. 4d, e), or in basal or isoproterenol-induced lipolysis (Fig. 4f, g). We additionally undertook transcriptional profiling of in vitro differentiated RSPO3-KD DFAT cells using RNA sequencing (RNA-seq) (Fig. 4h and Supplementary Fig. 7). Gene-set enrichment analysis identified positive regulation of cell death as the top biological process enriched in the cluster of genes differentially regulated in RSPO3-KD adipocytes (Fig. 4i). Consistent with these findings, doxycycline-induced RSPO3-KD rendered gluteal, but not abdominal, DFAT adipocytes more resistant to TNFα-induced apoptosis as determined by caspase-3/7 activity assays (Fig. 4j, k and Supplementary Fig. 8). Collectively, these findings indicate that in vitro RSPO3 modulates AP and adipocyte biology in a depot-dependent manner.

**RSPO3 regulates WNT signalling in APs**. We next examined if the biological activity of RSPO3 in APs was mediated by WNT signalling. RSPO3-KD in imAbdo and primary abdominal APs led to reduced WNT/β-catenin signalling as determined by decreased expression of the universal β-catenin target gene AXIN2 and reduced active β-catenin and phosphorylated LRP5/6 (pLRP5/6) protein levels, which are additional markers of canonical WNT pathway activation (Fig. 5a–c). In contrast, RSPO3-KD in imGlut and primary gluteal cells paradoxically led to increased canonical WNT signalling (Fig. 5a–c). We confirmed these results using RSPO3-KD imAbdo and imGlut cells stably expressing the TOPflash promoter reporter, which monitors endogenous β-catenin transcriptional activity (Fig. 5d). KD of LGR4 recapitulated the effects of RSPO3-KD on WNT/β-catenin signalling (Supplementary Fig. 9). Finally, we examined the consequences of RSPO3-KD on non-canonical WNT signalling (Fig. 5e, f). RSPO3-KD was associated with increased JNK phosphorylation, a marker of planar cell polarity WNT pathway activation, in abdominal APs and reduced CAMKII phosphorylation, a marker of WNT-calcium signalling, in gluteal cells. Thus, consistent with its depot-specific effects on AP function, RSPO3 activates distinct WNT pathways in abdominal and gluteal APs in vitro.

**A rspo3 mutation alters regional adiposity in zebrafish**. To verify that RSPO3 modulates AT biology in vivo we investigated a zebrafish line (sa14295) harbouring a mutation, which induces a premature stop codon in the final exon of rspo3 (Fig. 6a)[33]. We

chose to use this animal model as targeted disruption of Rspo3 is embryonic lethal in mice[34] and since Rspo3 expression is negligible in mouse mature adipocytes versus APs (Supplementary Fig. 10). Compared with their wild-type (wt) siblings, rspo3$^{sa14295}$ homozygous mutant (rspo3$^{m/m}$) animals had ~ 73% lower rspo3 expression in abdominal AT, encompassing the SC abdominal and visceral depots, and ~ 92% lower rspo3 expression in peripheral AT, comprising the SC AT on the lateral flank (Fig. 6b). Nile Red staining of AT-localised lipid revealed that rspo3$^{m/m}$ adults had significantly increased generalised adiposity versus wt siblings which, after adjustment for total body area, was only significant in females (Fig. 6c, d and Supplementary Fig. 11a–d). The enhanced adiposity in rspo3$^{m/m}$ females was driven by increases in the size of both abdominal and peripheral fat depots (Supplementary Fig. 11e, f) although peripheral AT expansion was more pronounced, leading to reduced abdominal-to-peripheral fat ratio (Fig. 6e and Supplementary Fig. 11g). rspo3$^{m/m}$ females also exhibited increased adipogenic (pparγ) gene expression in the abdominal depot (Fig. 6f). Finally, reduced rspo3 expression was associated with impaired proliferation, as evidenced by diminished EdU incorporation in abdominal adipose nuclei and adipocyte hypertrophy (Fig. 6g–i). Enhanced adiposity and adipocyte hypertrophy were confirmed in rspo3$^{m/m}$ juveniles prior to overt sexual differentiation (Supplementary Fig. 11h–n). In summary, Rspo3 inhibits AT expansion in vivo in zebrafish in a sex- and depot-specific manner.

## Discussion

The work described herein extends GWAS findings by demonstrating, through (1) co-localisation of the GWAS and adipose cis-eQTL signals, (2) in vitro functional studies in abdominal and gluteal APs and adipocytes and (3) in vivo adipose phenotyping of rspo3 mutant zebrafish, that RSPO3 is the effector gene and AT the target tissue of the WHRadjBMI associations at the RSPO3 locus. The demonstration that both rs72959041 and rs9491696 are additionally quantitative trait loci for plasma RSPO3 protein levels further supports this conclusion[35]. We also show that increased RSPO3 expression in SC fat is associated with impaired peripheral AT storage capacity and concomitant expansion of the android fat depots, in keeping with the insulin resistance phenotypes observed in carriers of WHRadjBMI-increasing alleles at this locus (Supplementary Table 1 and refs. [36,37]). Consistent with these data as well as our in vitro studies, increased RSPO3 mRNA abundance in SC adipocytes was also associated with increased gluteal adipocyte size in female subjects. Enlarged adipocytes have been shown to be associated with AT dysfunction, decreased adipogenic gene expression and systemic insulin resistance in humans[38–40]. Based on our gene

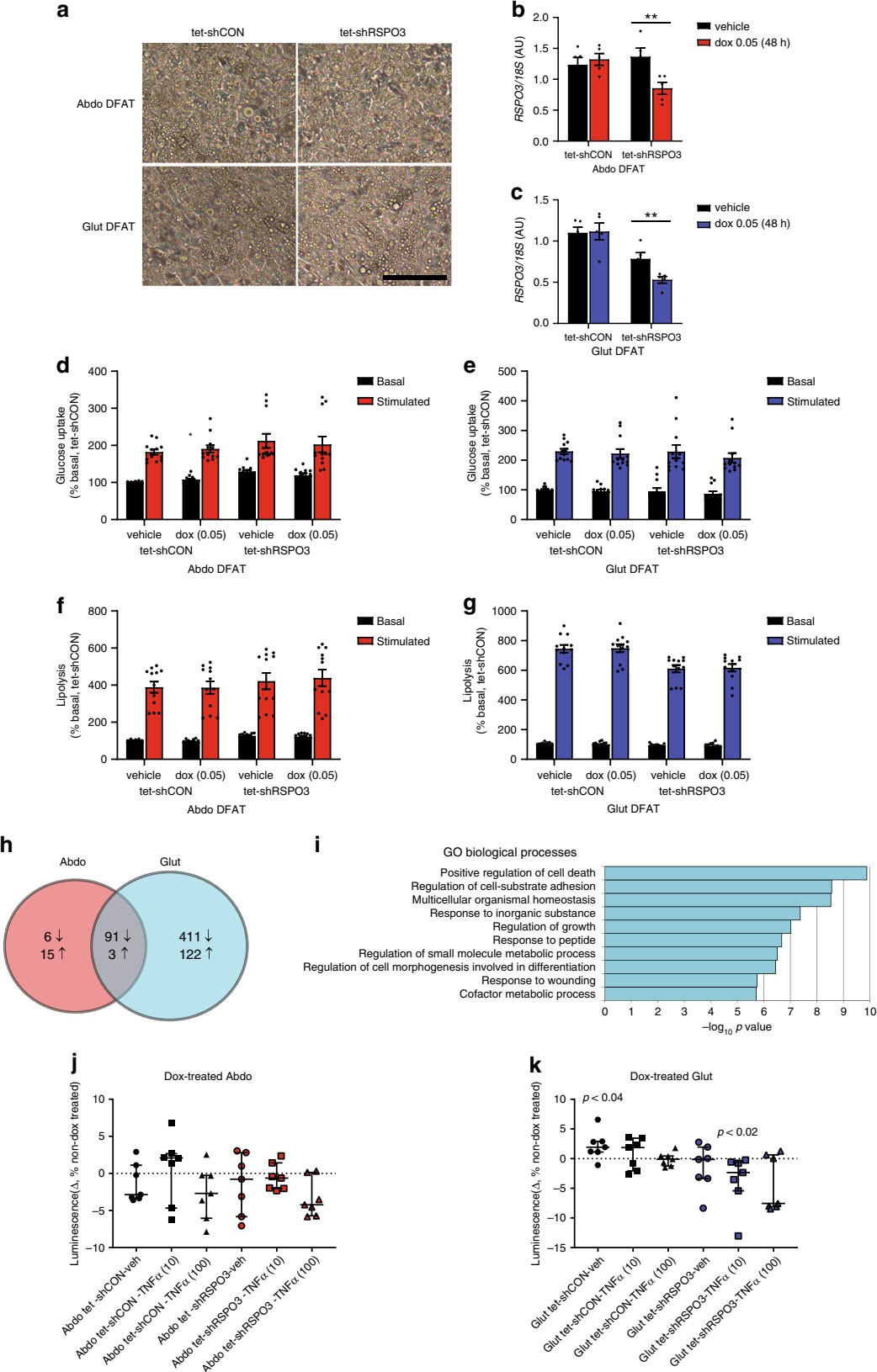

expression and AEI data we hypothesise that the sexually dimorphic pattern of WHRadjBMI associations observed at the RSPO3 locus, with stronger effects in females, stems from the lower baseline RSPO3 expression in female SC AT. Our AEI analyses and collective findings also support a model whereby

RSPO3 modulates regional adiposity via both autocrine and paracrine actions. The latter involve facilitating crosstalk between mature adipocytes and APs, in keeping with its mode of action in other organs and tissues (e.g., liver, gut, adrenal gland) where it functions as a local microenvironment-derived signal to modulate

**Fig. 4 Effects of doxycycline-induced *RSPO3*-KD on lipolysis, glucose uptake, and gene expression changes in in vitro differentiated DFAT cells. a** Light microscopy of abdominal and gluteal DFAT stable cell lines after 14 days adipogenic differentiation. Scale bar = 100 μm. **b, c** Expression of *RSPO3* in DFAT stable cell lines at day 15 of adipogenic differentiation following ~ 48-h treatment with 0.05 μg ml$^{-1}$ doxycycline versus vehicle in hormone-free basal media ($n = 5$ experiments). **d, e** Insulin-stimulated glucose uptake in DFAT cells relative to basal glucose uptake following ~ 48-h treatment with 0.05 μg ml$^{-1}$ doxycycline or vehicle in hormone-free basal media ($n = 12$, from four independent experiments). **f, g** Isoproterenol-stimulated glycerol release in DFAT cells relative to basal glycerol release following ~ 48-h treatment with 0.05 μg ml$^{-1}$ doxycycline or vehicle ($n = 12$, from four independent experiments). **b–g** Histogram data are means ± s.e.m. *$p < 0.05$, **$p < 0.01$. Statistical significance was assessed by two-tailed paired (**b, c**) and unpaired (**d–g**) Student's *t*-test comparing doxycycline versus vehicle treatment. **h, i** Transcriptional profiling of in vitro differentiated abdominal and gluteal DFAT cells following 48-h doxycycline-induced *RSPO3*-KD. **h** Number of differentially expressed (DE) genes. FDR $p < 0.05$. **i** Gene-set enrichment analysis results of DE genes in gluteal DFAT dox-induced *RSPO3*-KD cells showing the top 10 GO biological processes. **j, k** Effect of doxycycline-induced *RSPO3*-KD in differentiated **j** abdominal, and **k** gluteal cells on apoptosis in the presence of indicated concentrations of rhTNFα (ng ml$^{-1}$). Apoptosis was assayed using Caspase Glo 3/7 reagent. Results are shown as a % of luminescence of non-dox (vehicle)-treated cells in the presence of the same concentration of rhTNFα ($n = 7$ independent experiments). Statistical significance was assessed by a two-tailed Wilcoxon signed-rank test comparing dox and non-dox treated cells. Solid symbols, tet-shCON cells; open symbols, tet-shRSPO3 cells; circles, no rhTNFα; squares, 10 ng ml$^{-1}$ rhTNFα; triangles, 100 ng ml$^{-1}$ rhTNFα. Error bars are median values with interquartile ranges. Source data are provided as a Source Data file.

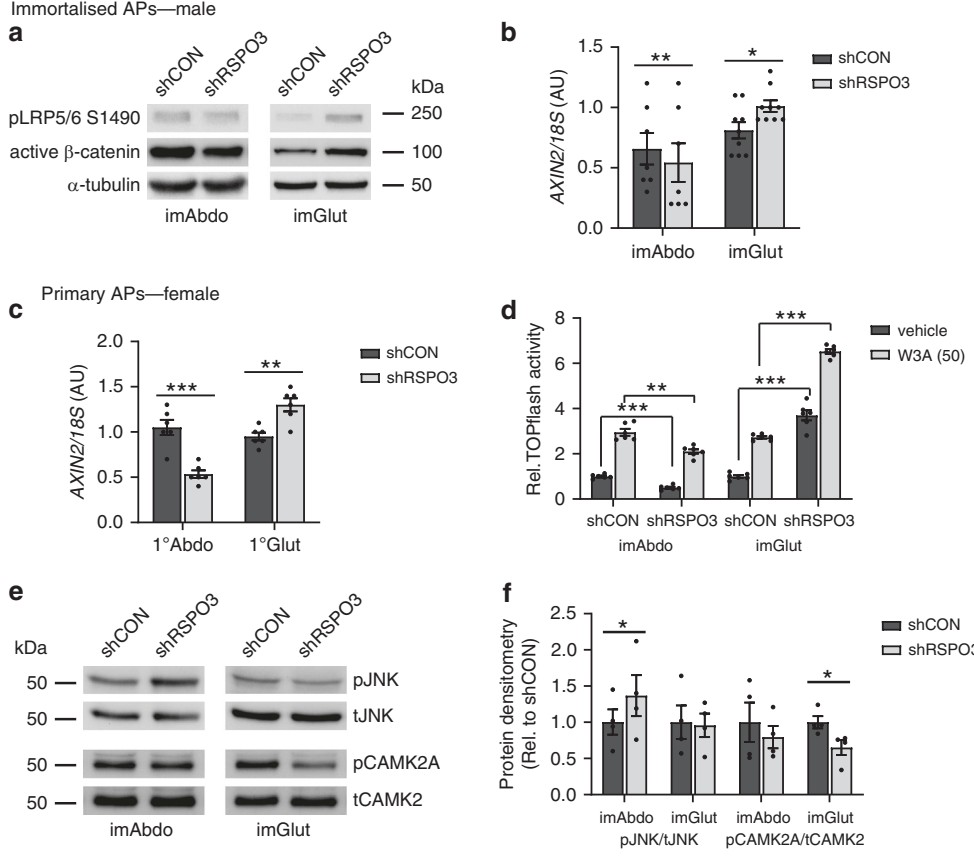

**Fig. 5 Effects of *RSPO3*-KD on canonical and non-canonical WNT signalling in SC abdominal and gluteal APs. a** Western blots of canonical WNT signalling markers and **b** qRT-PCR of AXIN2, a WNT/β-catenin target gene, in control (shCON) and RSPO3-KD (shRSPO3) imAbdo ($n = 7$ experiments) and imGlut APs ($n = 9$ experiments). **c** qRT-PCR of AXIN2 in shCON and shRSPO3 1° APs. **d** RSPO3-KD in imAbdo versus imGlut cells has differential effects on TOPflash promoter activity [W3A (50), 20-h treatment with 50 ng ml$^{-1}$ rhWNT3A] ($n = 6$ replicates/treatment). **e** Western blots of non-canonical WNT signalling markers in shCON and shRSPO3 imAbdo and imGlut APs. **f** Protein densitometry for pJNK/tJNK and pCAMK2A/tCAMK2 are shown relative to shCON levels. $n = 4$ independent experiments. Histogram data are means ± s.e.m. *$p < 0.05$, **$p < 0.01$, ***$p < 0.001$. Statistical significance was assessed by two-tailed paired (**b, c, f**) and unpaired (**d**) Student's *t*-test. Source data are provided as a Source Data file.

the growth, renewal and differentiation potential of stem and progenitor cells[41–44]. Indeed, in rodents, the AP activation that drives adipocyte hyperplasia in obesity was shown to be regulated by the adipose depot microenvironment[45].

Consistent with the distinct developmental origin of different fat depots and a role for RSPO3 in modulating fat distribution, AT RSPO3 mRNA abundance exhibited a central-to-peripheral gradient; visceral > abdominal > gluteal AT. SC AT RSPO3 expression was also higher in males than females in keeping with the increased male propensity to accumulate upper-body fat. Based on fractionated AT data from females, gluteal AP RSPO3 expression correlated selectively and negatively with lower-body fat accrual. In agreement with this finding, RSPO3-KD enhanced adipogenesis preferentially in gluteal progenitors in vitro.

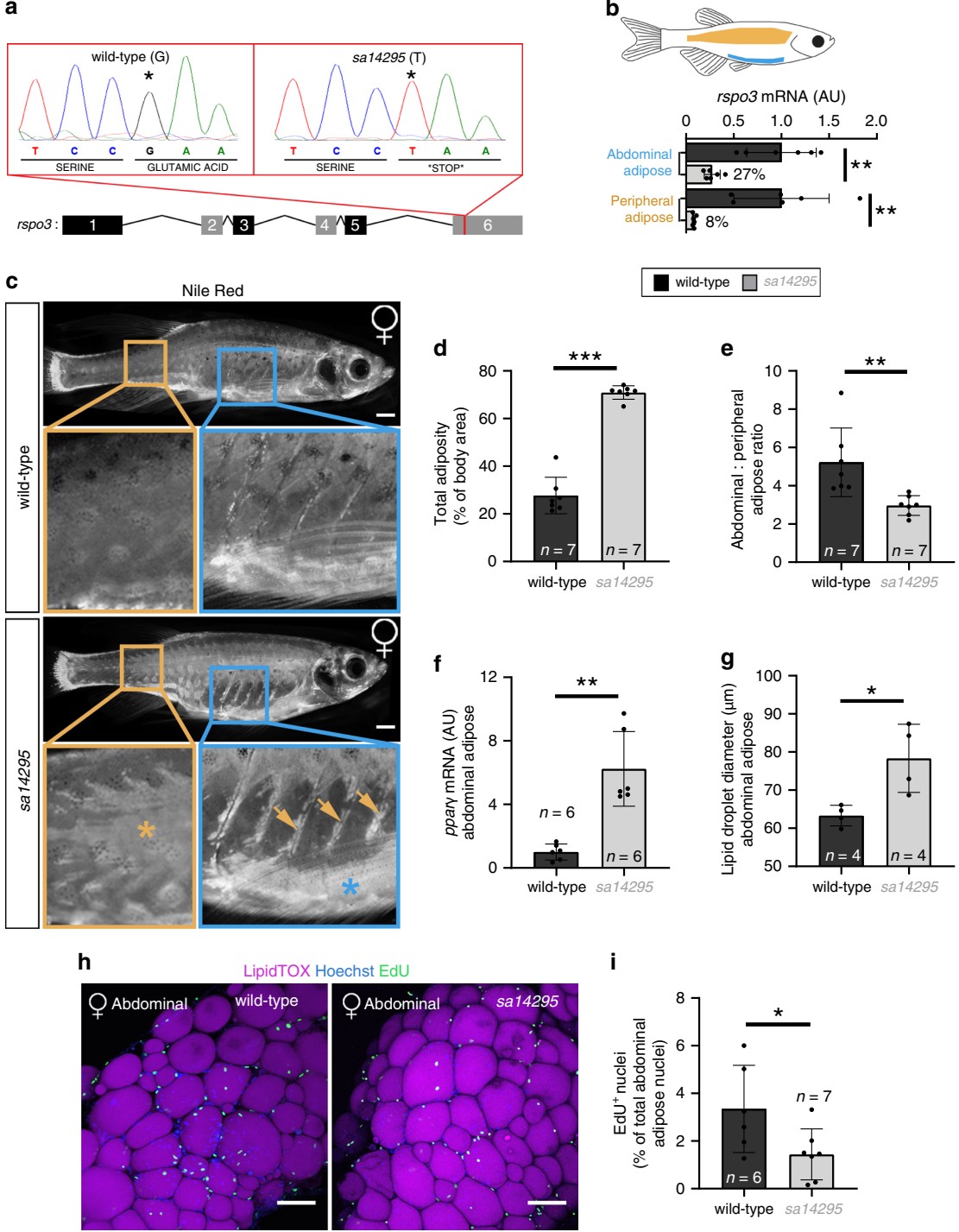

**Fig. 6 Effects of a nonsense *rspo3* mutation on total and regional adiposity in zebrafish. a** Schematic illustrating the location of the *sa14295* mutation in exon 6 of zebrafish rspo3. **b** qRT-PCR for rspo3 in abdominal and peripheral AT in wild-type sibling and *sa14295* homozygous (rspo3*m/m*) adult females (*n* = 6). **c** Nile Red staining of adult wild-type and rspo3*m/m* females (the blue asterisk indicates abdominal AT, whereas, the orange asterisk indicates peripheral AT that extends ventrally as marked by arrowheads). Scale bars = 1 mm. **d** Total adiposity, expressed as the % of total adipose-lipid area relative to body area, is significantly increased in rspo3*m/m* adult females. **e** The ratio of abdominal to peripheral SC AT is decreased in rspo3*m/m* adult females. **f** pparγ mRNA is elevated in the abdominal AT of rspo3*m/m* adult females. **g** Adipocyte-localised lipid droplets are significantly larger in rspo3*m/m* adult females. **h** Maximum intensity projection of female abdominal AT stained with LipidTOX (magenta), Hoechst (blue) and EdU (green). Scale bars = 100 μm. **i** Quantification showing decreased EdU+ nuclei in abdominal AT of rspo3*m/m* animals. Two-tailed unpaired Student's *t*-test was used for comparisons between genotypes (**b**, **d**–**g** and **i**). Histogram are means ± s.d. *p < 0.05, **p < 0.01, ***p < 0.001. Source data are provided as a Source Data file.

Directionally opposite results were observed in gain-of-function studies. The weaker effects of recombinant RSPO3 on AP biology in these latter experiments may be because RSPOs (and WNTs) are generally tightly associated with cell membrane and extracellular matrix proteoglycans, which are essential for efficient signalling[46]. Based on transcriptome-wide profiling and caspase-3/7 activity assays we speculate that RSPO3 may further restrain expansion of the gluteofemoral depot by increasing the susceptibility of gluteal adipocytes to apoptotic stimuli. We also hypothesise on the basis of our in vitro findings that RSPO3 may additionally promote centripetal fat deposition by stimulating expansion of the AP pool in the abdominal (and visceral) depots. In this regard, RSPO3 was shown to function as an organ size rheostat in the liver and gut by stimulating hepatocyte and intestinal progenitor cell proliferation respectively[18,47]. Animal studies have also demonstrated that AP number plays a critical role in determining the size of adult fat depots. Specifically, activation of PI3K-AKT signalling selectively in Myf5$^+$ mouse mesenchymal precursors was shown to lead to an increase in AP numbers and profound lipomatosis. Strikingly, because mouse ATs are composed of varying proportions of Myf5$^+$ and Myf5$^-$ progenitors, these animals displayed marked changes in fat distribution[48]. Recapitulating these findings, activating mutations in the PI3K-AKT pathway in humans lead to segmental AT overgrowth syndromes[49–51]. Finally, in vivo studies in mice have shown that PI3K-AKT2-induced AP proliferation is indispensable for hyperplastic AT growth during adulthood and contributes directly to the differential fat distribution between the sexes[45,52,53]. Notably, rare loss-of-function AKT2 mutations in humans are associated with partial lipodystrophy and severe insulin resistance[54].

The distinct biological responses elicited by RSPO3 in abdominal versus gluteal APs in vitro were driven by the capacity of RSPO3 to differentially modulate WNT signalling in these two cell types. While RSPO3-KD led to inhibition of WNT/β-catenin signalling in abdominal progenitors, the opposite was true in gluteal cells. Given that β-catenin is an oncogene, these findings are consistent with the diminished proliferative capacity selectively seen in RSPO3-KD abdominal APs. Interestingly, a recent study found that RSPO3 knockout led to increased proliferation in SGBS cells[55]; an AP line derived from a human infant with Simpson–Golabi–Behmel syndrome[56]. This provides further evidence that RSPO3 may have cell-type specific effects on AP proliferation and presumably WNT/β-catenin signalling. In contrast, the enhanced differentiation potential of RSPO3-KD gluteal cells may be driven by non-canonical WNT signalling since the WNT/β-catenin pathway is generally thought to inhibit adipogenesis. Consistent with this possibility RSPO3-KD was associated with inhibition of WNT-calcium signalling selectively in gluteal cells.

In keeping with the results of the human studies, a nonsense mutation in rspo3 in zebrafish was associated with increased generalised adiposity. Strikingly, changes in adiposity were more pronounced in female versus male adult fish consistent with the sexual dimorphism of the WHRadjBMI GWAS associations observed at the human RSPO3 locus[12]. Furthermore, this phenotype was independent of total body area, a reliable metric for body weight in wild-type zebrafish (see methods) and as such was likely to be driven by changes in AT biology rather than central nervous system effects on energy balance. We acknowledge that the close relationship between body area and body weight may be distorted in rspo3 mutants consequent to changes in body composition. Nonetheless, given that in absolute terms the increased adiposity in female mutants was ~3-fold higher than wild-type animals it is unlikely that any under- or over-estimation of body weight based on surface area would have altered this result.

rspo3$^{m/m}$ females also exhibited altered fat distribution in keeping with depot- and/or dose-dependent effects of Rspo3 on AT expansion and consistent with an earlier report demonstrating that rspo3 regulates anteroposterior patterning in zebrafish embryos[57]. Finally, this mutation was associated with impaired proliferation concomitant with adipocyte hypertrophy in abdominal AT. These data provide additional evidence that genetic mechanisms controlling regional adiposity are conserved between mammals and fish[58].

Our study has limitations. Specifically, APs undergo significant changes in gene expression when grown on tissue culture plastic and comprise a heterogeneous collection of cells thereby making it difficult to draw cell autonomous conclusions. For example, the pro-proliferative effect of RSPO3 in abdominal cells could have inhibited adipogenesis via non-cell autonomous effects (e.g., a non adipogenic cell out proliferating the adipogenic cells). Hence, it is uncertain whether the distinct biological responses elicited by RSPO3 in abdominal and gluteal cells in vitro are preserved in vivo and further studies, e.g., employing stable isotopes to determine in vivo cell turnover in the abdominal and gluteal AT of different RSPO3 genotype carriers are necessary to establish this. Additionally, we did not take into consideration the potential of eggs accounting for the increased body area in the female rspo3 mutant zebrafish. If this were true however, we have under- rather than over-estimated the increased body-fat percentage relative to body area of these animals. We also opted to express the body-fat data as a percentage of surface area rather than weight, to control for differences in zebrafish size.

In summary, we demonstrate that AT-specific regulation of RSPO3 mediates the WHRadjBMI genetic associations at the corresponding locus. Consistent with this finding, RSPO3 appears to have depot-specific effects on AP and adipocyte biology in vitro, which in APs are mediated at least partly, via modulation of WNT signalling. Our data reinforces the notion that developmental genes play an important role in hardwiring AP and adipocyte identity and support a role for RSPO3 as a secreted signal mediating crosstalk between adipocytes and APs to modulate AT expansion.

## Methods

**Study population, body-fat measurements and AT sampling.** The OBB (www.oxfordbiobank.org.uk) is a population-based cohort of 30–50-year-old healthy subjects, recruited from Oxfordshire in the UK, from whom basic anthropometric data were recorded and fasting blood samples taken for DNA and plasma chemistry. Menopause status of female subjects was determined by questionnaire. Perimenopausal females <51 years were assigned as pre-menopausal, and those ≥51 years as post-menopausal, based on 51 years being the average menopause age in the UK. Whole-body DXA was performed using a Lunar iDXA scanner (GE Healthcare, Little Chalfont, U.K.) and the acquired images processed using the enCORE v14.1 software. Paired SC abdominal and gluteal AT biopsies were obtained from adult subjects by needle biopsy from the periumbilical and buttock areas. Paired SC abdominal and visceral fat samples were obtained from patients undergoing elective surgery as part of the MolSURG study. All studies were approved by the Oxfordshire Clinical Research Ethics Committee (NRES Committee South Central-Oxford C, 08/H0606/107+5, IRAS project ID 136602), and all volunteers gave written, informed consent.

**Genotyping of OBB subjects.** OBB subjects were genotyped on the Illumina Human Exome BeadChip and Affymetrix UK Biobank Axiom arrays. Genotype imputation was performed using the Affymetrix UK Biobank Axiom array with Haplotype Reference Consortium (HRC), 1000Genome and UK10K reference panels using IMPUTE2 software.

**Adipose tissue histology and cell sizing.** AT biopsies were fixed in 10% neutral buffered formalin, embedded in paraffin wax, cut into 5 μm sections, and stained with haematoxylin and eosin. Sections were viewed and photographed at x200 magnification, and adipocyte cross-sectional area was measured using Adobe Photoshop 5.0.1 (Adobe Systems, San Jose, CA) and Image Processing Tool Kit (Reindeer Games, Gainesville, FL). Adipocyte volume and weight were calculated

for each individual using the following formulae:

$$V = \frac{\pi \times d^3}{6} \qquad (1)$$

$$w = V \times 0.915 \qquad (2)$$

$V$ = cell volume ($\mu m^3$), $d$ (real cell diameter) = histological cell diameter ($\mu m$) x 1.1, $w$ = weight of a single adipocyte (x$10^{-12}$ g), density of fat cell triglycerides = 0.915 g ml$^{-1}$. Adipocytes were assumed to be spheres.

**Cell lines.** Primary APs were separated from mature adipocytes by centrifugation following ~ 1-h collagenase (Roche)-digestion (1 mg ml$^{-1}$ in Hanks' buffered salt solution) of AT biopsies. Immortalised APs were generated by transgenesis of primary APs with human telomerase reverse transcriptase and HPV-E7 onco-protein. De-differentiated fat (DFAT) cells were derived from immortalised human AP cells as follows: AP cells were differentiated in the presence of 50 ng ml$^{-1}$ recombinant human BMP2, and 180 $\mu M$ fatty acid mix (oleate 75 $\mu M$, palmitate 65 $\mu M$, linoleate 40 $\mu M$ complexed to 10% BSA) was added from differentiation day 7 onwards. Culture media containing detached, well-differentiated, lipid laden cells were transferred to inverted 9 cm$^2$ slide flasks filled with standard growth media for ceiling culture, and allowed adhere and de-differentiate for 3–5 days. The flasks were then inverted and the media was replaced with fresh standard growth media to allow de-differentiated cells to proliferate. The resulting DFAT AP daughter cells have high adipogenic capacity.

**Cell culture and differentiation of human APs.** APs were grown in Dulbecco's modified Eagle's medium nutrient mixture F-12 Ham (DMEM-F12) supplemented with 10% fetal bovine serum (FBS), 2 mM L-glutamine, 0.25 ng ml$^{-1}$ fibroblast growth factor, 100 units ml$^{-1}$ penicillin and 100 $\mu g$ ml$^{-1}$ streptomycin. For differentiation of APs to adipocytes, confluent cells were cultured for 14–18 days in a standard adipogenic medium (DMEM-F12 containing 2 mM L-glutamine, 100 units ml$^{-1}$ penicillin, 100 $\mu g$ ml$^{-1}$ streptomycin, 17 $\mu M$ pantothenate, 100 nM human insulin, 10 nM 3,3',5-triiodo-L-thyronine, 33 $\mu M$ biotin, 10 $\mu g$ ml$^{-1}$ human transferrin and 1 $\mu M$ dexamethasone). For the first 4 days, 250 $\mu M$ 3-isobutyl-1-methylxanthine and 4 $\mu M$ troglitazone were added to the adipogenic medium. For quantitative measurement of intracellular lipid, APs were differentiated in type I collagen-coated 96-well plates, then assayed using the AdipoRed assay reagent (Lonza) and a CytoFluor Multi-well Plate Reader series 4000 (PerSeptive Biosystems) or an EnSpire 2300 Multilabel Reader (Perkin Elmer). To study the effect of recombinant human (rh) RSPO3 (3500-RS, R&D Systems) treatment on adipogenesis, confluent cells were cultured for 2 days in the presence of rhRSPO3 or vehicle, then differentiated throughout in the presence of rhRSPO3 or vehicle in standard adipogenic medium, or serum-containing adipogenic medium (DMEM-F12 containing 10% FBS, 2 mM glutamine, 100 units ml$^{-1}$ penicillin, 100 $\mu g$ ml$^{-1}$ streptomycin, 100 nM human insulin and 1 $\mu M$ dexamethasone), and assayed as above.

**Lentiviral constructs and generation of stable cell lines.** RSPO3 (shRSPO3, TRCN0000056663) and control shRNA plasmid vectors were purchased from Sigma-Aldrich. The LGR4 (shLGR4, TRCN0000004742) shRNA plasmid vector was purchased from Thermo Fisher Scientific. The 7xTcf-FFluc//SV40-mCherry (7TFC) TOPflash reporter lentiviral vector[59] was a gift from Roel Nusse (Addgene #24307). Lentiviral particles were produced in HEK293 cells (ATCC, CRL-1573) using MISSION® (Sigma-Aldrich) packaging mix. Stable cell-lines were generated by lentiviral transduction and selection in 2 $\mu g$ ml$^{-1}$ puromycin[26].

**Proliferation assays.** T75 or T175 flasks were seeded with equal numbers of control or KD cells. Cells were trypsinised and double counted using a haemocytometer or a Cellometer Auto T4 (Nexcelom Bioscience), every 96 h. Doubling time was calculated using the formula:

$$T_d = (t_2 - t_1) \times [\log(2) \div \log(q_2 \div q_1)] \qquad (3)$$

where $t$ = time (days), $q$ = cell number. To study the effects of rhRSPO3 treatment, cells seeded in 96-well plates were cultured in the presence of 5 ng ml$^{-1}$ rhRSPO3 or vehicle for up to 5 days. One plate of cells was assayed each day using CyQUANT Direct Cell Proliferation Assay and doubling time was calculated as above.

**Luciferase reporter assay.** To study the effects of RSPO3-KD on TOPflash activity, control and shRSPO3 cells expressing the 7TFC reporter vector[59] were grown to confluence in 96-well plates in complete growth media, then treated with indicated concentrations of rhWNT3A (5036-WN, R&D Systems) or vehicle in serum-free media for 20 h. To assess LGR4-KD on TOPflash activity, control and shLGR4 cells expressing 7TFC were treated with rhRSPO3 or vehicle in serum-free media for 24 h. TOPflash reporter activity was measured using the Luciferase Assay System (Promega) on a Veritas Microplate Luminometer (Turner Biosystems). Luciferase results were corrected to mCherry fluorescence intensity to adjust for differences in copy number of 7TFC in the different cell-lines.

**Doxycycline-inducible *RSPO3*-KD in adipocytes.** To study the effects of RSPO3-KD in in vitro differentiated adipocytes, oligonucleotides for shRSPO3 (top: 5'CCGGGCTGTGCAACATGCTCAGATTCTCGAGAATCTGAGCATGTTGCACAGCTTTTT; bottom: 5'AATTAAAAAGCTGTGCAACATGCTCAGATTCTCGAGAATCTGAGCATGTTGCACAGC), and shCON (top: 5'CCGGCAACAAGATGAAGAGCACCAACTCGAGTTGGTGCTCTTCATCTTGTTGTTTTT; bottom: 5'AATTAAAAACAACAAGATGAAGAGCACCAACTCGAGTTGGTGCTCTTCATCTTGTTG) were annealed and cloned into the tet-pLKO-puro doxycycline-inducible expression lentiviral vector [kind gift of Dmitri Wiederschain (Addgene #21915)][60]. DFAT preadipocyte cells stably transduced with the tet-pLKO-puro-shRSPO3 (tet-shRSPO3) or tet-pLKO-puro-shCON (tet-shCON) lentiviral vectors were maintained in tetracycline-free media [DMEM-F12 supplemented with 10% FBS (ThermoFisher Scientific Gibco, #26140079), 2 mM L-glutamine, 0.25 ng ml$^{-1}$ fibroblast growth factor, 100 units ml$^{-1}$ penicillin and 100 $\mu g$ ml$^{-1}$ streptomycin, and 2 $\mu g$ ml$^{-1}$ puromycin], and differentiated in standard adipogenic media. Differentiated cells were incubated in hormone-free basal media (DMEM-F12 supplemented with 2 mM L-glutamine, 100 units ml$^{-1}$ penicillin, 100 $\mu g$ ml$^{-1}$ streptomycin, 17 $\mu M$ pantothenate, 33 $\mu M$ biotin, 10 $\mu g$ ml$^{-1}$ human transferrin) containing either vehicle or 0.05 $\mu g$ ml$^{-1}$ doxycycline (to induce shRNA-expression) for 48 h prior to glucose uptake or lipolysis assays, or harvesting for RNA and protein.

**Glucose uptake assay.** Forty-eight hours before glucose uptake assay, 12-well plates of in vitro differentiated cells were starved in basal medium containing either vehicle or 0.05 $\mu g$ ml$^{-1}$ doxycycline to induce RSPO3-KD. On treatment day, cells were incubated in either fresh basal medium (to measure basal uptake) or basal medium containing 25 nM insulin for 30 minutes at 37 °C, 5%$CO_2$. Cells were then washed twice in HEPES-buffered saline (HBS; 140 mM NaCl, 20 mM HEPES, 5 mM KCl, 2.5 mM MgSO$_4$, 1 mM CaCl, pH7.4), incubated with 1 ml of uptake buffer [10 $\mu M$ 2-deoxy-D-glucose (2-DG) and 0.024 MBq (0.66 $\mu$Ci) per ml 2-[$^3$H]-DG in HBS] for 10 min at room temperature, washed twice in ice-cold 0.9% NaCl, and lysed in 1.2 ml 50 mM NaOH. Radioactivity was measured in liquid scintillant (Perkin Elmer) in a Beckman LS6500 Multipurpose Scintillation Counter (Beckman). To determine nonspecific diffusion, cells without insulin treatment were incubated with uptake buffer containing 10 $\mu M$ Cytochalasin B to block facilitative transport. Results were corrected for nonspecific diffusion, and normalised to protein concentration. All experiments were performed in triplicate and repeated at least three times.

**Lipolysis.** Basal and stimulated lipolysis was performed on in vitro differentiated cells in 12-well plates as follows: 2 days before lipolysis experiment, differentiated cells were incubated in basal media containing either 0.05 $\mu g$ ml$^{-1}$ doxycycline or vehicle. On treatment day, cells underwent a 2-h wash out period in Krebs Ringer HEPES (KRH) buffer containing 5 mM glucose and 3.5% bovine serum albumin (BSA), then incubated for 2 h in fresh KRH buffer (for basal lipolysis) or KRH buffer containing 10 nM isoproterenol (for stimulated lipolysis). At the end of the 2-h period, the treatment buffer was collected for glycerol measurements, and cells harvested in lysis buffer containing 1% IGEPAL CA-630, 150 mM NaCl and 50 mM Tris-HCl pH 8.0 for protein. Glycerol concentrations were measured using the GY105 enzymatic assay (Randox Laboratories Ltd) on an ILAB 650 clinical analyser (Instrumentation Laboratory UK) and normalised to cellular protein concentration.

**Caspase-Glo 3/7 assay for apoptosis.** Stably transduced DFAT cells were differentiated in 96-well plates. On day 13 of differentiation, cells were treated with 0.05 $\mu g$ ml$^{-1}$ doxycycline (or vehicle) in hormone-free basal media for 24 h, then a further 24 h with basal media containing 0, 10 or 100 ng ml$^{-1}$ rhTNF$\alpha$ (ThermoFisher Scientific Gibco, #PHC3015), in the presence of 0.05 $\mu g$ ml$^{-1}$ doxycycline or vehicle. Apoptosis was assayed using the Caspase-Glo 3/7 assay (Promega) and a Veritas Microplate Luminometer (Turner Biosystems).

**Zebrafish studies.** All zebrafish experiments conformed to the U.S. Public Health Service Policy on Humane Care and Use of Laboratory Animals, using protocols approved by the Institutional Animal Care and Use Committee of Duke University. Zebrafish were raised, fed, and housed as described[58]. The rspo3$^{sa14295}$ allele was generated by ENU mutagenesis. Twenty amino acids were truncated by the induction of the nonsense mutation. Animals were obtained from the Wellcome Trust Sanger Institute Zebrafish Mutation Project as F3 embryos after out-crossing to Hubrecht long fin wt zebrafish in the previous generation. F3 adults were subsequently out-crossed to the Ekkwill wt strain and F4 carriers inter-crossed for experiments. Quantitative real-time PCR (qRT-PCR) primers were forward, 5'-AGATGCTGCTCCTCATTGCT, and reverse, 5'-CTGGCCCCTGTTACACAGTT. The reverse primer was used for sequencing. Zebrafish experiments were conducted on adult (5-month-old) or postembryonic zebrafish (21 or 26 days post fertilisation) wt sibling or rspo3$^{m/m}$ fish as below. Zebrafish staging was conducted according to[61,62]. To control for body size and weight we utilised body area, which is an accurate proxy for zebrafish body mass (mass (mg) = 4327.4627 + 0.0004094*body area; $R^2$ = 0.95, $n$ = 41; data used from ref.[63]). We utilised

multiple linear regression to control for the potential confounding effect of body area on AT area measurements. Unless noted, no effect of body area on AT measures was found.

**Nile red staining of zebrafish adipose tissue**. Five-month-old adult and post-embryonic wt sibling or rspo3$^{m/m}$ fish were stained with Nile red and images were obtained on a Leica M205 fluorescence stereoscope using a GFP bandpass filter. Standard length, body area and whole-animal Nile Red signal intensity were quantified in ImageJ/FIJI (v1.50a). To assess regional adiposity, background signal was first subtracted from each image and Nile Red signal was segmented based on pixel intensity. Generalised adiposity was expressed as a % of body area; peripheral adiposity was assessed within a 400 μm$^2$ ROI at the lateral SC depot (LSAT); and abdominal AT was quantified using a 400 × 200 μm$^2$ ROI centred at the abdominal SC (ASAT) and pancreatic visceral (PVAT) depots.

**Confocal imaging of zebrafish adipose tissue**. Adult and postembryonic wt or rspo3$^{m/m}$ animals were housed in 200 ml water system supplemented with 25 μM EdU for 72 h. Representative abdominal (PVAT) and peripheral (LSAT) adipose tissues were then dissected and fixed overnight in 4% paraformaldehyde. The Click-iT imaging kit was used to detect EdU+ nuclei (Life Technologies, #C10338), LipidTOX to detect lipid droplets (Life Technologies, #H34475), and Hoechst 33342 to detect nuclei (Life Technologies, #H3570). Samples were mounted, and z-stacks collected on a Zeiss LSM780 confocal microscope. Images were processed, segmented, and quantified.

**RNA isolation and quantitative real-time PCR**. Total RNA was extracted from adipose tissue biopsies using TRIzol® reagent (Invitrogen) and cultured APs and isolated adipocytes using the RNeasy Mini Kit (Qiagen). cDNA was synthesised using the High Capacity cDNA Reverse Transcription kit (Applied Biosystems). qRT-PCR assays were performed using TaqMan gene expression assays. Expression values were calculated by the ΔCT transformation method (ΔCT = efficiency $^{[calibrator\ Ct\ -\ sample\ Ct]}$) and normalised to PPIA and PGK1 in the case of tissue samples, and 18S in the case of cultured cells and isolated adipocytes. Zebrafish AT samples were dissected and RNA extracted using the QIAzol lysis reagent (Qiagen, #79306) and the RNeasy lipid tissue mini kit (Qiagen, #74804). cDNA was synthesised using the SuperScript III kit (Invitrogen, #18080). qRT-PCR data were normalised to 18S. TaqMan assays and primers used are listed in Supplementary Table 10.

**Allele expression imbalance (AEI) studies**. AEI studies were performed as follows: Briefly, qRT-PCR was performed on paired cDNA and genomic DNA (gDNA) samples from individuals heterozygous for both rs9491696 or rs72959041 and their corresponding transcribed (tagging) SNV using a TaqMan allelic discrimination assay to the tagging SNV (Life Technologies). To generate a standard curve, qRT-PCR of three pairs of gDNA homozygous for each of the two alleles mixed at different ratios (30:70, 40:60, 50:50, 60:40 and 70:30) were performed concurrently. The standard curve was generated from the mean observed % of effect allele (EA), which tags the WHRadjBMI risk allele. The observed % EA from each experimental gDNA and cDNA sample was estimated from the standard curve. Under idealised conditions, the observed % EA from heterozygous gDNA would be 50%. Significant deviation in observed % of EA in cDNAs from that of gDNA indicates the presence of AEI in RSPO3 expression.

**Genetic credible sets**. For each distinct GWAS association signal, we calculated an approximate Bayes factor in favour of association on the basis of allelic effect sizes and standard errors from the approximate conditional analysis adjusting for all other index variants in the region. For the jth variant,

$$\Lambda_j = \sqrt{\frac{V_j}{V_j + \omega}} \exp\left[\frac{\omega \beta_j^2}{2V_j\left(V_j + \omega\right)}\right] \quad (4)$$

where $\beta_j$ denotes the estimated allelic effect (log-OR), $V_j$ the corresponding variance, and $\omega$ (=0.04)[64] the prior variance in allelic effects. We then calculated the posterior probability that the jth variant drives the association signal,

$$\pi_j = \frac{\Lambda_j}{\sum_k \Lambda_k} \quad (5)$$

We then constructed the 99% credible set of variants for each signal, by ordering all variants in descending order of their posterior probability of association and including variants until the cumulative posterior probability of association reached 0.99. We then calculated the number of variants and length of the genomic region covered by each 99% credible set.

**Co-localisation of WHRadjBMI signals and SC AT cis-eQTL**. To test the explicit hypothesis that the GWAS association signals for WHRadjBMI and the subcutaneous abdominal adipose tissue eQTLs at the RSPO3 locus are explained by the same causal SNVs at the respective independent signals, we applied a Bayesian test

implemented in the R package coloc[65]. We used default priors supplied by the coloc package (P1 = $1 \times 10^{-4}$, P2 = $1 \times 10^{-4}$, and P12 = $1 \times 10^{-5}$; prior probabilities for association in the GWAS dataset, the eQTL dataset, and both) and a threshold of PP3 + PP4 ≥ 0.99 and PP4/PP3 ≥ 5, a cutoff previously suggested[66], was used for "strong" evidence of co-localisation. Analysis was performed using published meta-analysed WHRadjBMI GWAS summary statistics[12] and subcutaneous abdominal adipose tissue eQTL results from the TwinsUK study (see below). For each association pair assessed for co-localisation, SNVs within 500 kb of the lead SNV were considered. We obtained the effect estimates for each independent association signal (for both GWAS and eQTL) by performing approximate conditional analysis using GCTA[67], adjusting for all other independent variants in the RSPO3 region. We used a reference sample of ~6,000 unrelated (pairwise relatedness < 0.025) individuals of white British origin, randomly selected from the UK Biobank, to model patterns of LD between variants.

**TwinsUK RSPO3 cis-eQTL analysis**. The TwinsUK samples were genotyped on a combination of Illumina chips, namely the HumanHap300, HumanHap610Q, 1M-Duo and 1.2MDuo Illumina arrays. Samples were imputed using the Haplotype Reference Consortium (HRC) reference panel (http://www.haplotype-reference-consortium.org), using Minimac 3 on the Michigan Imputation Server (https://imputationserver.sph.umich.edu). Genotype data were filtered to exclude SNVs with HWE P < $1 \times 10^{-6}$, imputation $R^2$ < 0.8, or a minor allele frequency (MAF) < 0.01. RNA-seq expression data from subcutaneous adipose tissue were available for 766 twins. RNA-seq data were generated, quantified and normalised[68]. Expression data were adjusted for family structure (family and zygosity) and 50 PEER factors using mixed effects models fitted using the lmer function from the lme4 package in R, and expression residuals used for ciseQTL analysis. Exon-level ciseQTL analysis at the RSPO3 gene was conducted within a 2 Mb window surrounding the transcription start site using the MatrixeQTL package in R version 3.5.0, which employs a standard additive linear model, with BMI, age and RNA extraction batch included as covariates.

**RNA-seq library preparation and analysis**. DFAT tet-shCON and tet-shRSPO3 cells were differentiated by standard differentiation protocol. On day 13 of differentiation, cells were treated with doxycycline (final concentration of 0.05 μg ml$^{-1}$), or vehicle, in hormone-free basal media, for 2 days. On day 15, cells were harvested for RNA. RNA-seq was performed on samples from three independent experiments. Total RNA purification and on-column DNase I-treatment were performed using the RNeasy Mini kit (Qiagen). RNA concentration was assessed using the Nano-Drop ND-1000 (Labtech) and RiboGreen (Invitrogen) on the FLUOstar OPTIMA plate reader (BMG Labtech), and RNA quality using the Agilent 2100 Bioanalyzer (Agilent) and the 2200 or 4200 TapeStation (Agilent, RNA ScreenTape). RNA integrity number (RIN) estimates for all samples were between 7 and 10. Library preparation, cDNA sequencing and bioinformatics analysis were performed at the Oxford Genomics Centre (Wellcome Trust Centre for Human Genetics, Oxford, UK). Differential gene expression analysis was performed with the edgeR package[69], with multiple testing correction using edgeR's default Benjamini-Hochberg method for controlling the false-discovery rate (FDR). Gene-set enrichment analysis of differentially expressed genes (FDR < 0.05) was performed in Metascape[70]. Full protocol detailed in Supplementary Methods.

**Western blotting**. Whole-cell lysates were prepared in ice-cold lysis buffer containing 50 mM Tris pH 8.0, 250 mM NaCl, 5 mM EDTA, 0.5% IGEPAL CA-630, 10 mM sodium fluoride, 1 mM sodium orthovanadate and protease inhibitors (Complete EDTA-free, Roche), and quantified using the Bio-Rad DC Protein assay kit. Equal amounts of protein were resolved by SDS-PAGE, transferred onto polyvinylidene fluoride membrane (Bio-Rad), and incubated with indicated primary antibodies according to the manufacturer's instructions, followed by the appropriate horseradish peroxidase-conjugated secondary antibodies (DAKO), and detection by enhanced chemiluminescence (GE Healthcare). Protein densitometry was performed using Image J. Antibodies were from: Cell Signalling Technology [phospho-LRP5/6-S1490 rabbit pAb (#2568, 1:1000), phospho-JNK (Thr183/Tyr185) rabbit pAb (#9251, 1:1000)]; Merck Millipore [active β-catenin (8E7) mouse mAb (05-665, 1:2000)]; Santa Cruz Biotechnology [total JNK mouse mAb (sc-7345, 1:500), phospho-CaMKIIα (Thr286) rabbit pAb (sc12886-R, 1:250), total CaMKII rabbit pAb (sc-9035, 1:500)]; Abcam [α-tubulin rabbit pAb (ab15246, 1:2000)]; and DAKO [horseradish peroxidase-conjugated secondary antibodies: mouse IgG goat pAb (P0447, 1:1000); rabbit IgG goat pAb (P0448, 1:2000)].

**Statistical analysis**. Statistical analyses for association studies between RSPO3 SNVs and OBB quantitative traits were performed using PLINK v.1.07 (pngu.mgh.harvard.edu/~purcell/plink/). All quantitative traits were log transformed and analysed with an additive linear regression model adjusting a priori for age, sex and % total fat mass. All other statistical analyses were performed using SPSS 22, SigmaPlot 14.0 or GraphPad 7.04. Evidence for between-group differences were assessed using a two-tailed paired or unpaired Student's t-test, as appropriate, unless otherwise specified.

## Reporting summary. Further information on research design is available in the Nature Research Reporting Summary linked to this article.

## Data availability

Full TwinsUK RNA-seq expression data from subcutaneous adipose tissue are available by direct application to TwinsUK. Full GWAS summary statistics from the UK Biobank meta-analysis for WHRadjBMI can be found on https://doi.org/10.5281/zenodo.1251813. The source data underlying Figs. 1b, d, e, g–m, 2, 3a, b, d–i, k–m, 4b–k, 5, 6b, d–g, i, Supplementary Figs. 1, 2, 3a–c, 4a, b, d–f, 5, 6, 7b–d, 8, 9, 10, 11a–c, e–g, i–l, n, and Table 2 are provided as a Source Data file. RNA-seq data that support the findings of this study have been deposited in GEO with the accession code GSE149294. All other relevant data are available from the corresponding authors upon reasonable request.

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

## Acknowledgements

We are grateful to the OBB volunteers, as well as the CRU nursing and technical staff. We also thank John Broxholme and Ben Wright of the Bioinformatics and Statistical Genetics Core at the Wellcome Centre for Human Genetics, University of Oxford. C.C. is funded by a British Heart Foundation Clinical Research Fellowship (FS/16/45/32359). We would also like to acknowledge funding support from the British Heart Foundation (PG/12/78/29862), Heart Research UK, the European Foundation for the Study of Diabetes, the National Institutes of Health (R01-DK093399), and the American Heart Association (11POST7360004 and 13POST16930097). J.E.N.M. is funded by a joint British Heart Foundation and University of Edinburgh Fellowship. C.L.G. is funded by Versus Arthritis (20000). Part of this work was conducted using the UK Biobank resource under application number 9161. M.I.McC. has been funded by the following grants: Wellcome 090532, 203141, 106130, 098381, 212259; NIDDK U01DK105535 and MRC M004422/1, L0201491/1, J010642/1. The TwinsUK study was funded by the Wellcome Trust and European Community's Seventh Framework Programme (FP7/2007-2013). The TwinsUK study also receives support from the National Institute for Health Research (NIHR)- funded BioResource, Clinical Research Facility and Biomedical Research Centre based at Guy's and St. Thomas' NHS Foundation Trust in partnership with King's College London. K.S.S. is supported by MRC Grants MR/L01999X/1 and MR/M004422/1. J.P.K. is funded by a University of Queensland Development Fellowship (UQFEL1718945) and a National Health and Medical Research Council (Australia) Investigator grant (GNT1177938). Bioinformatics analysis for RNA-seq experiments is supported by the Wellcome Trust [203141/Z/16/Z]. The views expressed in this article are those of the author(s) and not necessarily those of the NHS, the NIHR, or the Department of Health.

## Author contributions

Conceptualisation, C.C., F.K.; Methodology, N.Y.L., J.E.N.M., C.C.; Investigation, N.Y.L., J.E.N.M., M.V., C.C., M.J.N., J.P.K., M.T., K.E.P., A.M.; Writing–original draft, N.Y.L., C.C., J.E.N.M., J.F.R., F.K.; Writing–review & editing, all authors; Funding acquisition, C.C., F.K.; Resources, C.C., N.D., F.K., J.F.R., J.E-S.M., K.S.S., C.L.G., D.M.E., M.I.McC.; Supervision, C.C., F.K.

## Competing interests

C.C. and F.K. have received funding from NovoNordisk and Takeda. M.I.McC. was a Wellcome Senior Investigator and an NIHR Senior Investigator. M.I.McC. has served on advisory panels for Pfizer, NovoNordisk, Zoe Global; has received honoraria from Merck, Pfizer, NovoNordisk and Eli Lilly; has stock options in Zoe Global and has received research funding from Abbvie, Astra Zeneca, Boehringer Ingelheim, Eli Lilly, Janssen, Merck, NovoNordisk, Pfizer, Roche, Sanofi Aventis, Servier & Takeda. As of June 2019, M.I.McC. is an employee of Genentech, and holds stock in Roche.
