## [Peer Review File · Nature Communications]

Reviewers' comments:

Reviewer #1 (Remarks to the Author):

Loh et al present an overall excellent functional genomics investigation of SNPs near RSPO3, a candidate gene regulator of body fat distribution and diabetes risk. The authors have done fairly extensive work beyond the human genetics associations, including in vitro studies and generation of a zebrafish model that recapitulates predictions from the associated human phenotypes. The recapitulation of adipogenesis phenotypes (in culture) and fat phenotypes (in zebrafish) is the strength of the paper. In addition, while it is difficult to definitely state that RSPO3 is the causal gene, the authors accumulate fairly extensive orthogonal pieces of data in support of this contention. My primary criticisms are not with the veracity of any individual piece of data; however, I would favor being more circumspect about some of the mechanistic conclusions as discussed below in addition to specific questions/concerns raised about some of the results.

Although it does generally appear that certain depot-specific "gene signatures"—as the authors show in Figure S5—are preserved in AP culture, in our experience the changes that transpire when "APs" are placed in culture are much more dramatic than any persistent similarities. If the authors' had serial genome scales analyses (RNA-seq/Chip-seq) on fresh cells versus those in culture, for example, I think it much less likely that they could make the case that the depot-specific cell behavior in vitro is reminiscent of the in vivo biology. I'm therefore skeptical about conclusions drawn about the in vivo depot-specific cellular/molecular mechanisms, based on the activity of the respective APs in culture, without substantially more evidence. This issue is compounded by the recognition that APs are a heterogeneous collection of cells (e.g. see many recent single cell RNA-seq manuscripts, including Science paper by P. Seale group), making it difficult to draw cell autonomous conclusions. As such, I believe the paper would be strengthened—and I would be much more enthusiastic about it—if they dramatically tempered their statements about the depot specific mechanistic biology, at least with their current level of evidence.

There seem to be some discrepancies between the authors' findings on AP proliferation and the recently published work of the Cowan group in Circulation Research on RSPO3. I focus specifically on the proliferation effects of LOF, not just because of the discrepancy with the Cowan paper, but also because of the well-known (at least in preadipocytes) link between proliferation and adipogenesis, that is that inhibition of proliferation (e.g. mitotic clonal expansion) also tends to inhibit adipogenesis. Given that the AP population is heterogeneous and RSPO3 is also secreted, do the author's have proof that the cells that actually turn into adipocytes are the same cells that exhibit the proliferation effect? I recognize that this manuscript was not likely published at the time that the authors submitted the current paper, but at this point, I think it important that they reconcile their findings with the Circ Res paper. I also respectfully wonder how important the proliferation data is to their overall conclusions, particularly given that the differentiation data seems more robust.

While I like the fact that the LGR data is supportive and lends additional evidence that the LOF/GOF is specific, I wonder why the authors didn't merge the RSPO3 GOF/LOF experiments with LGR LOF. The attenuation of the RSPO3 effect by LGR LOF would strengthen the mechanistic specificity of their findings.

Particularly given that RSPO3 is a secreted protein, is there any evidence that the genetic variants are associated not just with effects on transcription (eQTL) but also on protein levels?

Reviewer #2 (Remarks to the Author):

Loh and colleagues report effects of RSPO3 variants on adipose progenitors and adipocytes to

influence body fat distribution. They evaluated two association signals for association with gene expression level in adipose depots, adipocyte size and insulin resistance. Knockdown and over-expression experiments reveal depot-dependent effects on adipogenesis and apoptosis that correspond to differences in WNT signaling. An *rspo3* nonsense mutation in zebrafish showed an effect on total and abdominal vs peripheral adipose tissue. Both the human and zebrafish analyses showed consistent differences between males and females. The results show that this gene is an effector gene at the GWAS locus and that the gene exhibits striking mechanistic differences in adipocyte biology between depots.

The manuscript is clearly written and the mechanistic results are robust to use of several cell and organism models and analyses, including multiple replicates per experiment.

Major comment:

1. The two association signals described are in LD with each other ($D'=1$), and the effect allele A of rs72959041 is always present on a haplotype with the effect allele G of rs9491696 (and of rs1936807). All association analyses and allelic imbalance analyses without considering the other signal may incorrectly attribute effects to the wrong variant. The effect size of rs72959041 on Table 1 can be two-fold larger than that of rs9491696, so analyses that report results for rs9491696 without considering rs72959041 could reflect the subset of individuals or haplotypes that carry the effect allele of rs72959041. The association analyses and allelic imbalance analyses from Table 1, most supplementary tables, Figure 1, and supplementary figure 1 should be repeated as haplotype analyses, or analyses of rs9491696 should be reported after excluding individuals carrying rs72959041.

Minor comments

1. The subject of the title is regulatory variants, but results are not specific for variants and most of the results show mechanistic effects
2. Page 7 refers to Extended Table 1 that does not exist
3. Table S1 is missing a p-value for fasting insulin adjusted for BMI. Including sample size and any differences by sex in these studies could provide further support to the conclusions.

Reviewer #3 (Remarks to the Author):

The Authors aimed to investigate the molecular, cellular and whole-body effects of WHRadjBMI-associated alleles at *RSPO3* especially regarding in regional fat distribution and cardiometabolic risk. They reported both human and animal studies. The study is interesting and adds valuable information.

1. "RSPO3 is expressed in a sex- and depot-specific manner in AT": the Authors, in order to study how *RSPO3* modulates fat distribution and adipocyte size, measured *RSPO3* mRNA in paired abdominal and gluteal fat biopsies from 200 adults. In Table S2, demo features of the population studied are reported. Mean age of the female group studied was 44.2 ± 0.5 (min 33, max 53). This clearly includes females in fertile age as well as others likely menopausal. Likelihood of inclusion of menopausal women was even higher in the subgroups undergoing DXA (Table S9). Data should be analyzed taking into account the reproductive status of the women involved, as the well-known effect that menopause has on fat distribution and abundance could influence the analysis.

2. In order to evaluate in zebrafish modifications in weight the Authors used body area, which they say is an accurate proxy for zebrafish body mass This is true for measurement of standard zebrafish populations. However, for accurate evaluation of possible qualitative alteration (e.g. fat) direct weight measurement and qualitative evaluation (e.g. with chemical carcass analysis) should

be provided.

3. Figure S11: the Authors state "Body area was slightly increased in *rspo3m/m* females relative to wild-types, but was unchanged between genotypes in males." Did the Authors take into consideration the presence of eggs in female fish, which can affect body area?

They report data on adult and juvenile Zebrafish. Which were the exact ages of the fish? What "prior to overt sexual differentiation" exactly mean? Zebrafish undergoes several changes in sexual differentiation before definite sexual maturation.

S11 D: pictures of males fish only are reported. As the increase in adiposity normalized on the % of the body area was significant only in females, would be more informative to report pictures of female fish.

4. Discussion: "phenotype was independent of total body area, a reliable metric for body weight in zebrafish" see point 2.

Erica Villa

We would like to thank the reviewers for their constructive comments. Our responses to the points they raise are outlined below:

Reviewers' comments:

Reviewer #1 (Remarks to the Author):

Loh et al present an overall excellent functional genomics investigation of SNPs near RSPO3, a candidate gene regulator of body fat distribution and diabetes risk. The authors have done fairly extensive work beyond the human genetics associations, including in vitro studies and generation of a zebrafish model that recapitulates predictions from the associated human phenotypes. The recapitulation of adipogenesis phenotypes (in culture) and fat phenotypes (in zebrafish) is the strength of the paper. In addition, while it is difficult to definitely state that RSPO3 is the causal gene, the authors accumulate fairly extensive orthogonal pieces of data in support of this contention. My primary criticisms are not with the veracity of any individual piece of data; however, I would favor being more circumspect about some of the mechanistic conclusions as discussed below in addition to specific questions/concerns raised about some of the results.

Although it does generally appear that certain depot-specific “gene signatures”—as the authors show in Figure S5—are preserved in AP culture, in our experience the changes that transpire when “APs” are placed in culture are much more dramatic than any persistent similarities. If the authors’ had serial genome scales analyses (RNA-seq/Chip-seq) on fresh cells versus those in culture, for example, I think it much less likely that they could make the case that the depot-specific cell behavior in vitro is reminiscent of the in vivo biology. I’m therefore skeptical about conclusions drawn about the in vivo depot-specific cellular/molecular mechanisms, based on the activity of the respective APs in culture, without substantially more evidence. This issue is compounded by the recognition that APs are a heterogeneous collection of cells (e.g. see many recent single cell RNA-seq manuscripts, including Science paper by P. Seale group), making it difficult to draw cell autonomous conclusions. As such, I believe the paper would be strengthened—and I would be much more enthusiastic about it—if they dramatically tempered their statements about the depot specific mechanistic biology, at least with their current level of evidence.

We agree with the reviewer, that based on our experiments we cannot be certain that the demonstrated, distinct biological responses elicited by RSPO3 in abdominal and gluteal cells *in vitro* function to mediate its effects on fat distribution *in vivo* in humans. To address this criticism, we have made several textual changes to temper our conclusions regarding the mechanistic basis through which RSPO3 modulates the size of the abdominal and gluteal fat depots. In addition, we have modified the title of the manuscript and highlighted this particular point as a limitation of our study in the revised Discussion.

There seem to be some discrepancies between the authors’ findings on AP proliferation and the recently published work of the Cowan group in Circulation Research on RSPO3. I focus specifically on the proliferation effects of LOF, not just because of the discrepancy with the Cowan paper, but also because of the well-known (at least in preadipocytes) link between proliferation and adipogenesis, that is that inhibition of proliferation (e.g. mitotic clonal expansion) also tends to inhibit adipogenesis. Given that the AP population is heterogeneous and RSPO3 is also secreted, do the author’s have proof that the cells that actually turn into adipocytes are the same cells that exhibit the proliferation effect? I recognize that this manuscript was not likely published at the time that the authors submitted the current paper, but at this point, I think it important that they reconcile their findings with the Circ Res paper. I also respectfully wonder how important the proliferation data is to their overall conclusions, particularly given that the differentiation data seems more robust.

The pro-proliferative effect of RSPO3 in abdominal APs was robust and reproducible and was detected in the presence of growth media containing 10% foetal bovine serum which is a potent growth stimulus. It is also in keeping with the pro-mitotic effects of canonical WNT

signalling which when aberrantly activated promotes cancer development. The discrepancy between the findings of this study and those of the recently published work from the Cowan group is most probably due to cell-specific effects of RSPO3 on cell proliferation (and presumably WNT/ β -catenin signalling), as indeed supported by our experiments and shown for other R-spondin family members¹. Furthermore, this could have arisen either due to the heterogeneity of the AP populations used in this study, as posited by the reviewer, or alternatively consequent to the origin of the SGBS cells (i.e. from an infant with Simpson–Golabi–Behmel syndrome) used in the Cowan paper compared to the adult origin of the APs used in our experiments.

Regarding the relevance of the proliferation data to our overall conclusions, in most organs and tissues studied, R-spondins function to promote the maintenance and proliferation of stem and progenitor cells. Furthermore, as highlighted in paragraph 2 of the Discussion, RSPO3 was shown to function as an organ size regulator in the liver, gut and adrenal gland by stimulating progenitor cell proliferation^{2, 3, 4, 5, 6, 7, 8}. We believe (and our data supports) that RSPO3 plays a similar role, at least in abdominal (and visceral) AT. In the revised Discussion we now provide further evidence from published studies, in addition to those cited previously, supporting an important role for AP proliferation in determining the size of fat depots and body fat distribution.

While I like the fact that the LGR data is supportive and lends additional evidence that the LOF/GOF is specific, I wonder why the authors didn't merge the RSPO3 GOF/LOF experiments with LGR LOF. The attenuation of the RSPO3 effect by LGR LOF would strengthen the mechanistic specificity of their findings.

We did consider performing epistasis experiments examining the functional interaction between RSPO3 and LGR4 in APs as suggested by the reviewer. However, subsequent to the AEI results, demonstrating that waist-to-hip ratio increasing alleles at the *RSPO3* locus increase *RSPO3* expression primarily in mature adipocytes, we decided to focus instead on elucidating the role of RSPO3 in adipocyte biology. A secondary reason why we opted against these studies, was the relative complexity of generating double knockdown cells (the single knockdown immortalised cells are puromycin and blasticidin resistant and the primary cells become senescent after two rounds of lentivirus infection and antibiotic selection) as well as, the weak biological effects of recombinant RSPO3 detected in APs.

Particularly given that RSPO3 is a secreted protein, is there any evidence that the genetic variants are associated not just with effects on transcription (eQTL) but also on protein levels?

We thank the reviewer for raising this point. Both rs72959041 and rs2489623, which is in high mutual linkage disequilibrium with rs9491696, have been associated with plasma levels of RSPO3 at GWAS significance (rs72959041: $\beta = 0.3805$, $p = 2.34 \times 10^{-12}$; rs2489623: $\beta = 0.27$, $p = 3.63 \times 10^{-28}$, $n = 3301$)⁹. This information is now included in the results section.

Reviewer #2 (Remarks to the Author):

Loh and colleagues report effects of RSPO3 variants on adipose progenitors and adipocytes to influence body fat distribution. They evaluated two association signals for association with gene expression level in adipose depots, adipocyte size and insulin resistance. Knockdown and over-expression experiments reveal depot-dependent effects on adipogenesis and apoptosis that correspond to differences in WNT signaling. An *rspo3* nonsense mutation in zebrafish showed an effect on total and abdominal vs peripheral adipose tissue. Both the human and zebrafish analyses showed consistent differences between males and females. The results show that this gene is an effector gene at the GWAS locus and that the gene exhibits striking mechanistic differences in adipocyte biology between depots.

The manuscript is clearly written, and the mechanistic results are robust to use of several cell and organism models and analyses, including multiple replicates per experiment.

Major comment:

1. The two association signals described are in LD with each other ($D'=1$), and the effect allele A of rs72959041 is always present on a haplotype with the effect allele G of rs9491696 (and of rs1936807). All association analyses and allelic imbalance analyses without considering the other signal may incorrectly attribute effects to the wrong variant. The effect size of rs72959041 on Table 1 can be two-fold larger than that of rs9491696, so analyses that report results for rs9491696 without considering rs72959041 could reflect the subset of individuals or haplotypes that carry the effect allele of rs72959041. The association analyses and allelic imbalance analyses from Table 1, most supplementary tables, Figure 1, and supplementary figure 1 should be repeated as haplotype analyses, or analyses of rs9491696 should be reported after excluding individuals carrying rs72959041.

Thank you. We have now highlighted in the relevant Results section that rs72959041 and rs9491696 are in LD and adjusted all analyses of rs9491696 for rs72959041 genotype.

Minor comments

1. The subject of the title is regulatory variants, but results are not specific for variants and most of the results show mechanistic effects

The title of the manuscript has been revised to reflect this.

2. Page 7 refers to Extended Table 1 that does not exist

Thank you for pointing this out. We had uploaded the Excel file containing Extended Table 1 incorrectly in our original submission. This has now been amended.

3. Table S1 is missing a p-value for fasting insulin adjusted for BMI. Including sample size and any differences by sex in these studies could provide further support to the conclusions.

Thank you. The missing p-value has been added to Supplementary Table 1. As per the reviewer's advice the sample size of the studies quoted has now also been added to the Table. Finally, we now provide in the relevant Results section of the revised manuscript data from the Magic consortium (Lagou et al., unpublished) demonstrating that rs1936807 is associated with sexually dimorphic effects on fasting insulin levels which are stronger in women; consistent with the associations of this signal with fat distribution.

Reviewer #3 (Remarks to the Author):

The Authors aimed to investigate the molecular, cellular and whole-body effects of WHRadjBMI-associated alleles at RSPO3 especially regarding in regional fat distribution and cardiometabolic risk. They reported both human and animal studies. The study is interesting and adds valuable information.

1. "RSPO3 is expressed in a sex- and depot-specific manner in AT": the Authors, in order to study how RSPO3 modulates fat distribution and adipocyte size, measured RSPO3 mRNA in paired abdominal and gluteal fat biopsies from 200 adults. In Table S2, demo features of the population studied are reported. Mean age of the female group studied was 44.2 ± 0.5 (min 33, max 53). This clearly includes females in fertile age as well as others likely menopausal. Likelihood of inclusion of menopausal women was even higher in the subgroups undergoing DXA (Table S9). Data should be analyzed taking into account the reproductive status of the women involved, as the well-known effect that menopause has on fat distribution and abundance could influence the analysis.

Thank you. As advised by the reviewer we have revised Table 2 after adjusting all data to menopausal status (as well as, age and percent total fat mass). After doing so, the results remained essentially unchanged. We have not reanalysed the expression quantitative locus data because the allelic qPCR experiments demonstrated evidence of allelic imbalance for both SNVs of interest regardless of menopausal status.

2. In order to evaluate in zebrafish modifications in weight the Authors used body area, which they say is an accurate proxy for zebrafish body mass This is true for measurement of standard zebrafish populations. However, for accurate evaluation of possible qualitative alteration (e.g. fat) direct weight measurement and qualitative evaluation (e.g. with chemical carcass analysis) should be provided.

We opted to express the body fat data as a percentage of surface area rather than weight, to control for differences in zebrafish size. The reviewer is right to point out that the close relationship between weight and surface area may be distorted in animals with mutations affecting body composition. Unfortunately, we do not have weight (or chemical carcass analysis) measurements and as we currently don't maintain this line it would not have been possible to generate these data within the timeframe allocated for the revision. However, given that in absolute terms the increased adiposity of female mutants was approximately 3-fold higher than wild-type animals, it is unlikely that any under- or over-estimation of body weight based on surface area would have altered this result.

In response to the reviewers comment the relevant paragraph of the discussion has been revised as follows: 'In keeping with our *in vitro* studies, a nonsense mutation in *rspo3* in zebrafish was associated with increased generalised adiposity. This phenotype was independent of total body area, a reliable metric for body weight in **wild-type** zebrafish (see methods) and as such was **primarily likely to be driven** by changes in AT biology rather than central effects on energy balance. **We acknowledge that the close relationship between body area and body weight may be distorted in *rspo3* mutants consequent to changes in body composition. Nonetheless, given that in absolute terms the increased adiposity in female mutants was approximately 3-fold higher than wild-type animals it is unlikely that any under- or over-estimation of bodyweight based on surface area would have altered this result.**'

Additionally this point is now highlighted in the revised Discussion as a potential limitation of our study.

3. Figure S11: the Authors state "Body area was slightly increased in *rspo3*m/m females relative to wild-types, but was unchanged between genotypes in males." Did the Authors take into consideration the presence of eggs in female fish, which can affect body area?

We did not take into consideration the potential of eggs accounting for the increased body area in the female *rspo3* mutants. If this were true however, we have under- rather than over-estimated the increased body fat percentage relative to body area of these animals. This point is now addressed in the revised Discussion.

They report data on adult and juvenile Zebrafish. Which were the exact ages of the fish? What "prior to overt sexual differentiation" exactly mean? Zebrafish undergoes several changes in sexual differentiation before definite sexual maturation.

The juvenile fish were 21 and 26 days post fertilisation. We now cite two references (^{10, 11}) for standard length and post-embryonic stages based on which we determined the developmental progression of zebrafish in the Methods section of the paper. We have taken care to report these results in context of previous descriptions by us and others of the

close relationship between body size and adipose tissue development in the zebrafish (e.g. see ^{10,11}).

S11 D: pictures of males fish only are reported. As the increase in adiposity normalized on the % of the body area was significant only in females, would be more informative to report pictures of female fish.

As requested by the reviewer images of Nile Red stained adult females are now provided in Fig 6C.

4. Discussion: "phenotype was independent of total body area, a reliable metric for body weight in zebrafish" see point 2.

This was addressed in our response to comment 2.

References

1. Wu C., *et al.* RSPO2-LGR5 signaling has tumour-suppressive activity in colorectal cancer. *Nat Commun* **5**, 3149 (2014).
2. Planas-Paz L., *et al.* The RSPO-LGR4/5-ZNRF3/RNF43 module controls liver zonation and size. *Nat Cell Biol* **18**, 467-479 (2016).
3. Rocha A. S., *et al.* The Angiocrine Factor R-spondin3 Is a Key Determinant of Liver Zonation. *Cell Rep* **13**, 1757-1764 (2015).
4. Sigal M., *et al.* Stromal R-spondin orchestrates gastric epithelial stem cells and gland homeostasis. *Nature* **548**, 451-455 (2017).
5. Greicius G., *et al.* PDGFRalpha(+) pericryptal stromal cells are the critical source of Wnts and RSPO3 for murine intestinal stem cells in vivo. *Proc Natl Acad Sci U S A* **115**, E3173-E3181 (2018).
6. Hilkens J., *et al.* RSPO3 expands intestinal stem cell and niche compartments and drives tumorigenesis. *Gut* **66**, 1095-1105 (2017).
7. Basham K. J., *et al.* A ZNRF3-dependent Wnt/beta-catenin signaling gradient is required for adrenal homeostasis. *Genes Dev* **33**, 209-220 (2019).
8. Harnack C., *et al.* R-spondin 3 promotes stem cell recovery and epithelial regeneration in the colon. *Nat Commun* **10**, 4368 (2019).
9. Sun B. B., *et al.* Genomic atlas of the human plasma proteome. *Nature* **558**, 73-79 (2018).
10. Imrie D., Sadler K. C. White adipose tissue development in zebrafish is regulated by both developmental time and fish size. *Dev Dyn* **239**, 3013-3023 (2010).
11. Minchin J. E. N., Rawls J. F. A classification system for zebrafish adipose tissues. *Dis Model Mech* **10**, 797-809 (2017).

REVIEWERS' COMMENTS:

Reviewer #1 (Remarks to the Author):

The authors have done a reasonable job tempering some their conclusions. The strength of the manuscript remains the human genetics component and the support for that signal provided by a slew of functional genomics studies. My enthusiasm for seeing the manuscript in Nature Communications stems from this aspect of the work.

In my opinion, the cell biological/mechanistic conclusions of the manuscript remain the weaker component. Their conclusions about depot specific mechanisms rest on immortalized cell preparations from a couple of human subjects and could very well be a manifestation of cell culture, immortalization, etc. Even within these cells, they have not carefully disentangled heterogeneity, proliferation, and differentiation. For example, could the pro-proliferative effect inhibit adipogenesis via non-cell autonomous effects (e.g. a non adipogenic cell out proliferating the adipogenic cells). This potential source of in vitro artifact remains a real possibility as proliferation and differentiation often travel together in adipogenesis (dependence on mitotic clonal expansion). There are no new experiments and therefore my original questions remain. Collectively, I remain skeptical about the leaps taken from the cell culture assays on immortalized cells from a couple of subjects, whereby RSPO3 mediates two disparate depot-specific cellular mechanisms in vivo, leading to essentially opposite effects on local adiposity.

As I signaled in my original review, however, I think the work is important enough that with sufficient acknowledgement of limitations of the in vitro studies, I would not object to publication.

Reviewer #2 (Remarks to the Author):

My comments were addressed.

Reviewer #3 (Remarks to the Author):

The Authors have satisfactorily answered the comments

REVIEWERS' COMMENTS:

Reviewer #1 (Remarks to the Author):

The authors have done a reasonable job tempering some of their conclusions. The strength of the manuscript remains the human genetics component and the support for that signal provided by a slew of functional genomics studies. My enthusiasm for seeing the manuscript in Nature Communications stems from this aspect of the work.

In my opinion, the cell biological/mechanistic conclusions of the manuscript remain the weaker component. Their conclusions about depot specific mechanisms rest on immortalized cell preparations from a couple of human subjects and could very well be a manifestation of cell culture, immortalization, etc. Even within these cells, they have not carefully disentangled heterogeneity, proliferation, and differentiation. For example, could the pro-proliferative effect inhibit adipogenesis via non-cell autonomous effects (e.g. a non adipogenic cell out proliferating the adipogenic cells). This potential source of in vitro artifact remains a real possibility as proliferation and differentiation often travel together in adipogenesis (dependence on mitotic clonal expansion). There are no new experiments and therefore my original questions remain. Collectively, I remain skeptical about the leaps taken from the cell culture assays on immortalized cells from a couple of subjects, whereby RSPO3

mediates two disparate depot-specific cellular mechanisms in vivo, leading to essentially opposite effects on local adiposity.

As I signaled in my original review, however, I think the work is important enough that with sufficient acknowledgement of limitations of the in vitro studies, I would not object to publication.

Thank you. We have now made additional textual changes to the manuscript to place greater emphasis to the limitations of the in vitro studies. In addition, we have further tempered our conclusions, based on the aforementioned studies, with regards to RSPO3 mediating two disparate depot-specific cellular mechanisms in vivo.

Reviewer #2 (Remarks to the Author):

My comments were addressed.

Reviewer #3 (Remarks to the Author):

The Authors have satisfactorily answered the comments